# Evapotranspiration Stress Intensifies with Enhanced Sensitivity to Soil Moisture Deficits in a Rapidly Greening China

Yuan Liu[1], Yong Wang[1,2], Yong Zhao[1,2], Shouzhi Chen[3], Longhao Wang[4,5], Wenjing Yang[6], Xing Li[1], Xinxi Li[7], Huimin Lei[8], Huanyu Chang[9], Jiaqi Zhai[1,2], Yongnan Zhu[1,2], Qingming Wang[1,2], and Ting Ye[10]

[1]State Key Laboratory of Simulation and Regulation of Water Cycle in River Basin, China Institute of Water Resources and Hydropower Research (IWHR), Beijing, 100038, China
[2]Key Laboratory of Water Safety for Beijing-Tianjin-Hebei Region of Ministry of Water Resources, China Institute of Water Resources and Hydropower Research (IWHR), Beijing, 100038, China
[3]College of Water Sciences, Beijing Normal University, Beijing, 100875, China
[4]Key Laboratory of Water Cycle and Related Land Surface Processes, Chinese Academy of Sciences, Beijing, 100101, China
[5]Institute of Geographic Sciences and Natural Resources Research, University of Chinese Academy of Sciences, Beijing, 100049, China
[6]National Centre for Groundwater Research and Training, Flinders University, Adelaide, SA 5001, Australia
[7]PowerChina Asia & Pacific, Power Construction Corporation of China, Beijing, 100038, China
[8]State Key Laboratory of Hydroscience and Engineering, Department of Hydraulic Engineering, Tsinghua University, Beijing, 100084, China
[9]Academy of Eco-civilization Development for Jing-Jin-Ji Megalopolis, Tianjin Normal University, Tianjin, 300387, China
[10]State Key Laboratory of Water Resources and Hydropower Engineering Science, Wuhan University, Wuhan, 430072, China

*Correspondence to*: Yong Wang (wangyong@iwhr.com)

**Abstract.** Amidst drastic environmental changes, the intricate interplay and feedback mechanisms in the water-vegetation-atmosphere nexus experience alteration. Previous research primarily centers on the responses among variables within this system, with little known about whether and how these responses (sensitivities) change. Here, we employ the Evapotranspiration Stress Index (ESI) to represent the equilibrium of the nexus and develop a memory dynamic linear model based on Bayesian forward filtering. The model takes into account the carry-over effect in the "dry gets drier" self-amplify loop, allowing for a more effective estimation of the ESI time-varying sensitivity to associated influencing factors. Our analysis reveals that from 1950 to 2020, mainland China experienced a notable 4.74% escalation in evapotranspiration stress. Surface soil moisture sever as the primary driver, whose sensitivity to ESI surged by 9.49%in the last decade compared to the early 2000s. Vapor Pressure Deficit (VPD) and Leaf Area Index (LAI) also exerted a substantial role, with their sensitivities fluctuating approximately 22.91% and -45.77%, respectively. Moreover, the greening pace is linked to an increase in soil moisture sensitivity and a decrease in VPD sensitivity, suggesting that rapid greening may alter the ecological resilience against soil deficit and atmospheric drought. In comparison, the widely used moving window multiple linear regression (MLR) significantly overestimates sensitivity fluctuations, necessitating prudent interpretation of numerical

estimates in related research findings. Our findings underscore the spatiotemporal variations in sensitivity, enriching the comprehension of ecosystem reactions to external factors, and offer essential insights for advancing greening endeavors.

**Graphical Abstract**

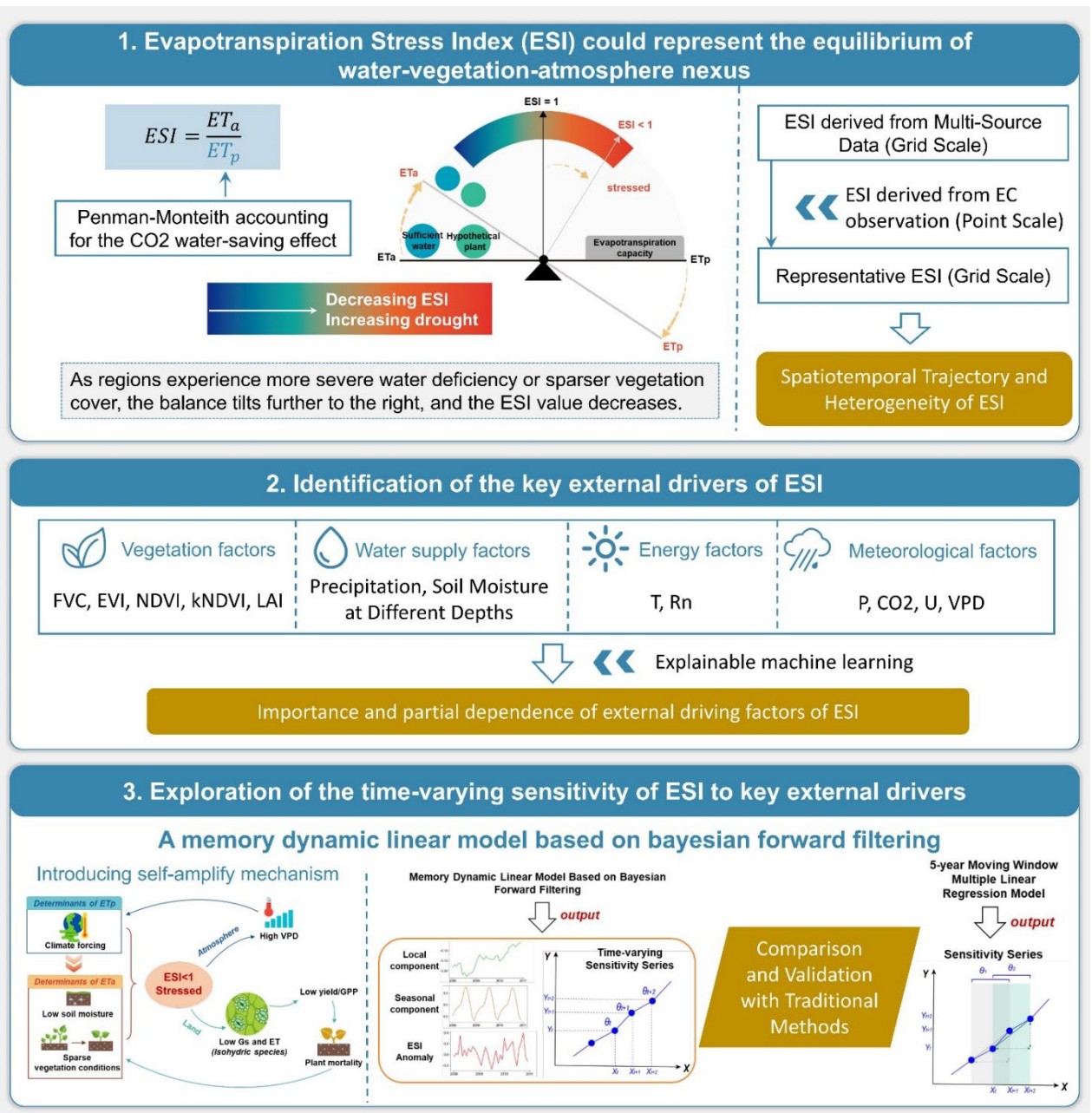

## 1 Introduction

Over the last two decades, approximately one-third of the global land has experienced vegetation greening(Chen et al., 2019). Concurrently, 53% to 64% of these regions have undergone atmospheric desiccation(Yuan et al., 2019). However, research has also revealed a rise in available water resources (precipitation supply minus evapotranspiration loss) in ~45% proximate and downwind zones(Cui et al., 2022). Contrary to greening, satellite-based studies have indicated a 0.4-fold expansion in regions shifting from greening to browning(De Jong et al., 2013; Liu et al., 2023). Amidst the ongoing debates over greening or browning, and wetting or drying, it is unequivocal that the intricate interactions and feedback mechanisms among water, vegetation, and climate are altered.

Defined as the ratio of actual to potential evapotranspiration, the Evapotranspirative Stress Index (ESI) can comprehensively reflect the equilibrium between water supply, vegetation status, and climatic conditions(Anderson et al., 2013, 2016) (Fig. 1ab). Potential evapotranspiration (ETp, also known as $ET_0$, reference crop evapotranspiration in agriculture) represents the evaporation potential under ideal circumstances—ample water supply, uniform plant growth, and consistent crop coverage (specifically, alfalfa)—and generally considered to be influenced solely by climatic factors (Allen et al., 1998; Li et al., 2022b; Thornthwaite, 1948). Regions suffering from water scarcity or limited vegetation cover are subject to evapotranspirative stress (Fig. 1b), whereas ample water or vigorous vegetation may result in actual evapotranspiration surpassing the hypothetical potential (Liu et al., 2022) (Fig. 1a). This occurs because, during the calculation of ETp, parameters such as canopy reflectance coefficient and surface resistance are fixed for a hypothetical crop, whereas actual vegetation characteristics can exceed those assumptions. Previous research has proven the the high sensitivity of ESI to early-stage drought, as plant naturally curtails its water consumption before drought fully manifest (Nguyen et al., 2019; Otkin et al., 2018). Therefore, comprehending the underlying mechanisms and associated factors of ESI dynamics is crucial for implementing prompt and effective drought mitigation strategies, protecting stable ecosystems, and maintaining high yields.

The evolution of evapotranspiration (ET) stress includes a positive "dry gets dryer" land-atmosphere feedback (Fu et al., 2022; Gentine et al., 2019; Seneviratne et al., 2010) (Fig. 1c). Regions subjected to climate-induced ET stress are characterized by low soil moisture and weak vegetation vitality. Isohydric species, in response, will close their stomata to minimize water loss, albeit at the cost of decreased photosynthesis and diminished carbon accumulation (Grossiord et al., 2020; Joshi et al., 2022). On one side, this reduction in water vapor release into the atmosphere intensifies atmospheric dryness, further reinforcing climate constraints. On the other side, sustained stress may cause plant mortality due to hydraulic failure and carbon starvation, resulting in progressively sparser terrestrial vegetation coverage (Kono et al., 2019; Mantova et al., 2022). Such a self-amplify loop highlights the necessity to consider temporal connections or sustained effects when exploring the ET stress dynamics. Research based on data analysis and field experiments has uncovered that such internal memory effects are particularly pronounced in water-limited areas (Liu et al., 2018; Richard et al., 2008; Xiao et al., 2024a). Statistically, the autocorrelation component can serve as a good representation of internal memory effects (Forzieri et al.,

2022; Smith et al., 2022). For example,  Kusch et al., (2022)employed Normalized Difference Vegetation Index (NDVI) lagged by one time step as a representation of intrinsic-vegetation memory and compared its role with external meteorological factors in resisting adverse environmental disturbance. Theoretically,  an increase in the autocorrelation coefficient suggests a decline in ecosystem resilience to external interventions or an approach to critical thresholds (Liu, 2019; Scheffer et al., 2009).It is disappointing that existing research primarily concentrated on external driving factors, such as meteorological, hydrological, and biotic drivers (Feng et al., 2020; Fu et al., 2024b; Jung et al., 2010; Liu et al., 2020c; Peng et al., 2019).Therefore, integrating the co-regulation of external forcing factors with internal memory effects is essential for an in-depth investigation into the dynamics of ET stress and ecological responses.

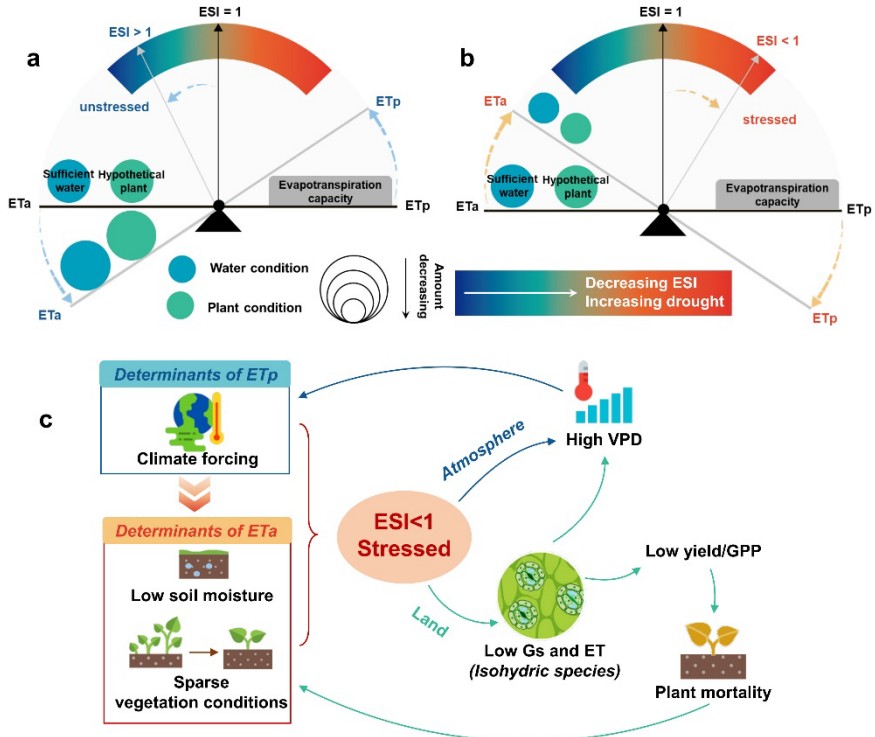

**Figure 1. Conceptual diagram of evapotranspiration stress and its self-amplify loop.** (a) ESI represents the hydrological-vegetation-atmosphere equilibrium, defined as the ratio of actual evapotranspiration (ETa) to potential evapotranspiration (ETp). When using the parameters of crops assumed in the PM equation to calculate ETp (also known as ET0, reference crop evapotranspiration in agriculture), ESI values greater than 1 may occur. (b) As regions experience more severe water deficiency or sparser vegetation cover, the balance tilts further to the right, and the ESI value decreases. A smaller ESI value indicates more severe drought conditions.  (c) Schematic of the self-amplify feedback loop, where Gs represents stomatal conductance, GPP stands for gross primary productivity, and VPD denotes vapor pressure deficit. Blue lines depict the atmospheric feedback pathway, while green lines represent the terrestrial surface process pathway.

Differences in regional evaporation stress patterns arise from different response to relevant factors. Soil moisture acts as the supply side of evaporation, while vapor pressure deficit (VPD), as the demand side, pulls water from soil pores and plant stomata into the atmosphere (Dong et al., 2020; Liu et al., 2020b). As an alleviator, increased ambient $CO_2$ can mitigate

evaporation by enhancing water use efficiency (Li et al., 2023; Liu et al., 2017). Among these external drivers, mounting evidence emphasizes the pronounced sensitivity to moisture and VPD (Liu et al., 2020a; Sulman et al., 2016; Zhang et al., 2021a; Zhong et al., 2023). However, several studies also argue that vegetation structure, characterized by the Leaf Area Index (LAI), plays a key role in controlling the distribution of surface energy and land-atmosphere coupling that determines evapotranspiration (Fu et al., 2022). In addition to regional characteristic dependence, the differing sampling time spans are also important contributors to the inconsistent results. In an ever-changing world, the sensitivity of evaporation stress to environmental changes may also evolve (Hsu and Dirmeyer, 2023). However, research on the time-varying sensitivity is still in its infancy, even the state-of-the-art Earth System Models have not yet incorporated it into parameterization schemes. Existing research investigates these temporal changes by segmenting time series. For instance, Zeng et al., (2022)conducted a regression analysis with 5-year moving window to derive the time series of sensitivity coefficients, which represent the NDVI's response to precipitation. Similarly, Li et al., (2022) applied interpretable machine learning algorithms to compute the sensitivity of LAI to soil water per 3-year-block data during 1982 to 2017 to investigate its temporal changes. However, these approaches only yield average response for each segment, rather than the true "instantaneous or dynamic" sensitivity. Fortunately, the application of Dynamic Linear Model based on Bayesian forward filtering, as demonstrated by Zhang et al., (2022) in their recent research of global NDVI sensitivity to precipitation, offers a promising avenue to elucidate this problem.

Over the past two decades, China's mean air temperature has increased at a rate of 0.24°C per decade (decade$^{-1}$), exceeding the global average of 0.20°C decade$^{-1}$. Concurrent with climate warming and large-scale ecological restoration initiatives— including the Three-North Shelter Forest Program, Grain for Green Program, and Plain Greening Project (Fu et al., 2024a), the regional leaf area index (LAI) has risen by 7.66% (4.21% decade$^{-1}$, p < 0.001), the highest rate globally (Chen et al., 2019). However, surface soil moisture has simultaneously declined at 0.08% decade$^{-1}$, signaling hydrological trade-offs. These rapid environmental transformations, driven by both climate variability and anthropogenic activities, have established China as a critical natural laboratory for studying coupled hydro-ecological-climatic interactions (Bai et al., 2020; Li et al., 2018; Zheng et al., 2022)

Here, a memory dynamic linear model (MDLM) that integrates intrinsic lagged effects and concurrent extrinsic forcing to quantify both the response of the evapotranspiration stress index (ESI) to environmental changes and its temporal variability. Utilizing the MDLM, we calculate the time-varying sensitivity of ESI to key drivers at pixel scale in China over the past two decades, with 8-day temporal granularity. The robustness of the results is verified through a moving-window multiple linear regression model. Prior to model implementation, the interpretable machine learning algorithms leveraging two classifiers are employed to screen key drivers of ESI across diverse climate-vegetation zones at 8-, 16-, and monthly timescales. This study aims to (a) untangle the dynamic trajectory of ET stress, (b) investigate the potential driving mechanisms of ESI, and (c) quantify the time-varying sensitivity of ESI to principal external drivers upon the incorporation of intrinsic memory effects. Our findings are expected to narrow the knowledge gap in temporal variability of evapotranspiration stress response,

enhance our comprehension of the complex water-vegetation-climate interactions, and offer insights into potential improvements for Earth System Models.

## 2 Materials and Methods

### 2.1 Datasets

The Evapotranspiration Stress Index (ESI) (Eq. 1) is calculated as the ratio of actual evapotranspiration (ETa) to potential evapotranspiration (ETp). Although ETp lacks of consensus on definitions and estimation methods, the Penman-Monteith (PM) equation is adopted here due to its consistency with the ESI framework. The ERA5-derived ETp product satisfies this criterion, delivering integrated datasets of radiation, surface temperature, and wind speed parameters required by the PM equation. To avoid potential overestimation of ETp, we implement the Yang et al., (2019)-modified PM formula, which

incorporates the $CO_2$ water-saving effect and is widely adopted in evapotranspiration studies (Lian et al., 2021; Zhang et al., 2023). $CO_2$ data were obtained from CarbonTracker (CT2022) for the period 2001–2020 at a 3° × 2° spatial resolution. Appendix A details the calculation procedures of ESI. Additionally, we assess the efficacy of other three mainstream evaporation products in representing ESI spatial patterns across China, benchmarking against observational data from 26 flux stations.

$$ESI = \frac{ET_a}{ET_p} \tag{1}$$

In addition to the meteorological and $CO_2$ data utilized in ETp calculations (Eq. A4), total precipitation (Prec), atmospheric pressure (P), and soil volumetric water content (svm) across different soil layers from ERA5 are also considered as influencing factors for the ESI. The root zone of vegetation serves as a critical interface for water-plant-atmosphere interactions (Gao et al., 2014, 2024; Wang-Erlandsson et al., 2016). Utilizing root zone depth estimations based on an 80-

145 year drought return period from Stocker et al. (2023), we computed the spatial distribution of root-zone soil moisture (Rsvm) across China through a weighted averaging algorithm, as detailed in Appendix B. Vegetation factors, which serve as key drivers of the Evaporative Stress Index (ESI), were incorporated through the Global Land Surface Satellite (GLASS) datasets, specifically the Leaf Area Index (LAI; version V60) and Fractional Vegetation Cover (FVC; version V40) (Liang et al., 2013, 2014, 2021), These datasets feature an 8-day temporal resolution and 0.05° spatial resolution spanning 2001–2020.

Additionally, Moderate Resolution Imaging Spectroradiometer (MODIS) datasets—including the Normalized Difference Vegetation Index (NDVI) and Enhanced Vegetation Index (EVI) (MOD13C1) with a 16-day temporal resolution—were integrated into the analysis. Besides, kernel density estimation is utilized to smooth the NDVI data, resulting in a kernel NDVI (kNDVI) that is believed to mitigate saturation effects and enhance robustness to noise (Camps-Valls et al., 2021). (Eq. 2).

$$kNDVI = \tanh(NDVI^2) \tag{2}$$

Grids with NDVI value below 0.1 are omitted to exclude areas of barren, rock, sand (i.e. deserts) or snow. Moreover, the Aridity Index (AI), is calculated as multiyear average (2001–2020) by the ratio of annual Prec to ETp using the ERA5 dataset. The lower the AI value, the drier the region, which is used to classify the study area into different hydrometeorological zones: sub-humid ($0.5 \leq AI < 0.65$), semi-arid ($0.2 \leq AI < 0.5$), and arid ($0.05 \leq AI < 0.2$). Regions with an AI exceeding 0.65 are categorized as non-dryland areas (Spinoni et al., 2015). To mitigate the effects of land-use changes, the annual China Land Cover Dataset (Yang and Huang, 2021), featuring a 30-meter grid spacing for the years 2001–2020, are employed. Only grids with consistent land cover types—cropland, forest, or grassland—over the entire study duration were selected for further analysis, as depicted in Fig A1.

The datasets and pre-processing procedures are summarized in Table S1 in Supplement. All variables were interpolated to a spatial resolution of 0.1° to match the ERA5 dataset and were composited over three distinct temporal scales: 8-day, 16-day, and monthly. The selection of datasets was guided by criteria that included high accuracy, fine spatial resolution, and a broad temporal coverage.

## 2.2 Identification of the key drivers

Regarding hydrothermal conditions and vegetation variables, there are several ways to represent each, with a varying number of proxies. For example, vegetation health can be gauged though multiple indicators such as FVC, LAI, NDVI, and kNDVI. In contrast, the energy conditions considered in this study are limited to T and Rn. A disparity in the number of variables may lead to a bias within the model, favoring certain feature groups and thereby undermining the accuracy of feature importance assessments (Liu et al., 2024). More critically, a high degree of correlation between features—known as multicollinearity—may cause overfitting, where the model becomes excessively tailored to the training data, compromising its ability to generalize. Before model construction, it is essential to identify the principal factors influencing ET stress. Explainable machine learning (SHapley Additive exPlanations) offers an efficacious approach to feature selection (Estécio Marcílio Júnior and Eler, 2020; Li et al., 2022a). We utilize explainable machine learning models, specifically XGBoost and Random Forest (RF) regressors, to assess the relative importance of drivers. SHAP values can indicate the marginal contributions of each predictor, with higher SHAP values denoting greater importance (Lundberg and Lee, 2017). Both XGBoost and RF, as ensemble-based approaches, mitigate overfitting risks and bolster result stability, yet they differ in their suitability for various scenarios, data volume, task complexity, and computational efficiency (Breiman, 2001; Chen and Guestrin, 2016). To optimize the regression model, we use grid search techniques and 5-fold cross-validation for parameter refinement. Furthermore, regularization parameters were integrated alongside the "Early Stopping" mechanism to mitigate overfitting risks and rigorously control model complexity. Moreover, we employ partial dependence plots (PDPs) to elucidate the effects of key drivers on the ESI, illustrating the model's average predictive response to changes in individual drivers, with other variables held constant (Friedman, 2001; Štrumbelj and Kononenko, 2014). PDPs provide a visual depiction of the dynamic interplay between ESI predictions and changes in critical factors. To enhance the robustness of PDPs, we performed 30 iterations with varying random seeds and averaged the resulting estimates.

Persistent common trends and analogous seasonal patterns can result in overfitting, thereby concealing the actual causal relationships (Li et al., 2022a) (Fig. 2a). In our SHAP model, we input de-seasonalized and detrended anomalies of each variable. Seasonality is mitigated by subtracting the climatological mean, and long-term trends are eliminated through the application of a locally weighted smoothing filter (LOWESS) with a span of 0.3 (Cleveland, 1979).

## 2.3 Exploration of the time-varying sensitivity

The sensitivity of evaporative stress to key drivers (identified in Sect. 2.2) in this study is a proxy of the response of the ESI to variations in drivers. Given the presence of self-amplify mechanisms, the previous state of evaporative stress can be represented by a lag-5 autocorrelation model. The mathematical expression for sensitivity is as follows:

$$\delta ESI_t = \theta_{ESI1}\delta ESI_{t-1} \sim \theta_{ESI5}\delta ESI_{t-5} + \theta_{Hydr}\delta Hydr_t + \theta_{Ener}\delta Ener_t + \theta_{Clim}\delta Clim_t + \theta_{Vege}\delta Vege_t + \varepsilon \quad (3)$$

where $\delta Hydr_t$, $\delta Ener_t$, $\delta Clim_t$ and $\delta Vege_t$ are anomalies in hydrological, energy, climatic, and vegetation conditions, respectively, with subscripts indicating that the anomalies are calculated from the current time(t). The corresponding θ represents the sensitivity of evaporative stress investigated in this study, and ε is the error term. $\theta_{ESI1}\delta ESI_{t-1} \sim \theta_{ESI5}\delta ESI_{t-5}$ denotes the stress states from the previous 1 to 5 stages. Considering the lagging effect, we constructed a memory dynamic linear model based on Bayesian forward filtering (MDLM) to analyze the time-varying sensitivity of ESI. This model draws upon the foundational work of Liu et al. (2019) and Zhang et al. (2021), which primarily examines the vegetation's response to antecedent growth states and prevailing climatic conditions.

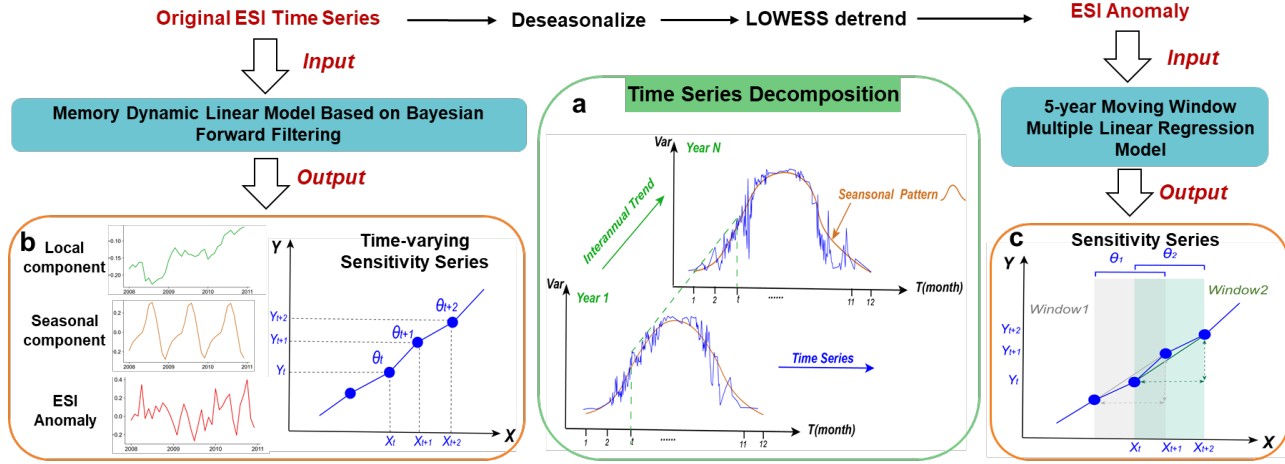

**Figure 2. Illustration of time series decomposition and procedures for sensitivity analysis.** (a) Conceptual diagram of the time series structure, where the blue line denotes the intra-annual sequence, the green line signifies the interannual trend, and the brown line illustrates the seasonal cycle pattern. (b) The output of the MDLM model. (c) The results from the 5-year moving window multiple linear regression model, in which Y represents the anomaly of ESI, X denotes the driving factors, θ symbolizes the sensitivity of Y to changes in X, and the subscript denotes the specific time point.

Dynamic linear models serve as a sophisticated statistical methodology for modeling time-series signals by incorporating various contributing factors (Prado et al., 2021). For each pixel, a MLDM was developed, encompassing long-term evaporation stress trends, seasonal cycles, the effects of internal memory from the previous five stages, and various external

drivers. DLM effectively decomposes the ESI series into three components: local mean and trend, seasonal pattern, and regression part. The MDLM, utilizing Bayesian forward filtering, enables the prediction of the dynamic relationship between ESI's regression components and driving factors at each timestep, depicting an "instantaneous" response. Additionally, this approach allows sensitivity analysis in both stationary and non-stationary scenarios (Liu, 2019; Zhang et al., 2021b), with model precision increasing as sample size grows. Therefore, we calibrated MLDM using an 8-day resolution, the finest

temporal scale available, with data allocation comprising 80% for training and 20% for validation. Further details on the MDLM model are available in Text S1 of the Supplement.

Furthermore, we employed a multiple linear regression (MLR) with a 5-year moving window to calculate a rough sensitivity sequence for comparison. To ensure the sensitivity reflects the relationship between the ESI anomalies and their drivers, we manually subtracted the long-term average within an 8-day interval from the original ESI series to remove seasonality.

Subsequently, we applied the LOWESS method to eliminate local trends, thereby obtaining the ESI anomalies. Based on Formula 3, we established a MLR model and analyzed the regression coefficients, namely, the sensitivity coefficients, across each moving window. It is crucial to acknowledge that the sensitivity derived from the MLR model represents an average response across a designated window (Fig. 2c). The MLR provides only a rough sequence of sensitivity, which is utilized to confirm the overarching trend throughout the study period.

The regression coefficients obtained by the two methods are partial regression coefficients, reflecting the magnitude and direction of the relationship between the variable and the ESI while holding other independent variables constant to remove the obfuscating effects of collinearity (Toyoda, 2024). The precise elucidation of real-world phenomena's intricacies necessitates incorporating a broad spectrum of representative drivers. To prevent overfitting due to multicollinearity, we evaluated the variance inflation factor (VIF) for the independent variables before modeling (Belsley, 1991) (Table S4 in

Supplement).

Besides, we employed Sen's slope estimator (Sen, 1968) to quantify the temporal trend and the Mann-Kendall (MK) test to ascertain the statistical significance (Kendall, 1949; Mann, 1945). The slope>0 signifies a positive trend and slope<0 denotes a negative trend over time.

This study comprehensively investigates the dynamic trajectory of evaporative stress by comparing the Evaporative Stress

Index (ESI) calculated using the conventional PM formula from 1950 to 2020 with the ESI derived from the PM formula incorporating $CO_2$ water-saving effects from 2001 to 2020, enabling a detailed examination of its variation characteristics. Subsequent analyses are conducted using the latter due to limitations in the influencing factors' data series. The SHAP model is applied at 8-day, 16-day, and monthly scales to obtain temporally robust importance rankings. Furthermore, to calculate sensitivity, the MDLM model operates at an 8-day temporal resolution, as a larger sample size enhances parameter accuracy.

The temporal evolution patterns of sensitivity coefficients are explored at an interannual scale.

## 3 Results

### 3.1 Evapotranspiration stress is intensifying in China

China mainland demonstrated a pronounced decline in the ESI from 1950 to 2020 (total change of 4.74%, decreasing by 0.54% $d^{-1}$, $p < 0.001$). Despite the elevated ESI values accounting for the $CO_2$ water-saving effect, the downward trend remains largely parallel to the long-term series, indicating a continued exacerbation of evaporative stress (-0.70% decade$^{-1}$, $p = 0.31$). Note that tracking the tendency annually over a 70-year time series confers enhanced statistical significance due to the expanded sample size (Fig. 3a). Collectively, the ESI for each month is on a downward trend. It remains stable during the initial phase of the growing season (0.58±0.08). Commencing in April, the onset of the growing season prompts a marked increase in ESI, coinciding with improved vegetation conditions, and culminates in a peak between 0.9 and 1.0 by August (Fig. 3b). Spatially, over one-quarter (28.01%) of the regions nationwide have registered a significant reduction in ESI ($p < 0.05$), concentrating in the North China Plain and the Northeast where the rate surpassing 2.40% decade$^{-1}$. Concurrently, stress in the Northwest and the middle to lower Yangtze River basin is easing (Fig. 3c). ESI exhibits a sharper decline in dryland (total change of 7.63%, decreasing by 0.91% decade$^{-1}$, $p < 0.001$) compared to the stable and gradual decrease in non-dryland lands (total change of 2.92%, decreasing by 0.32% decade$^{-1}$, $p < 0.01$) (Fig. 3d, S2). This decline is observed universally across various land use categories, with cropland experiencing the most substantial reduction and concurrently exhibiting the highest percentage of areas aggregated (-1.14% decade$^{-1}$, $p < 0.001$; $7.59 \times 10^5$ km$^2$, 50.24% of the total cropland area) (Fig. 3ce, S1, S2). Similar dynamics have been observed in ESI factoring in the water conservation benefits attributed to $CO_2$. Even so, it is alarming that the decline in ESI for drought-prone and agriculture-intensified regions in the recent two decades has outpaced the rate documented over the preceding 70 years (-1.63% decade$^{-1}$ vs -0.91% decade$^{-1}$ in dryland, -1.60% decade$^{-1}$ vs -1.14% decade$^{-1}$ in cropland). Consequently, the progression of water stress conditions in these areas necessitates increased scrutiny.

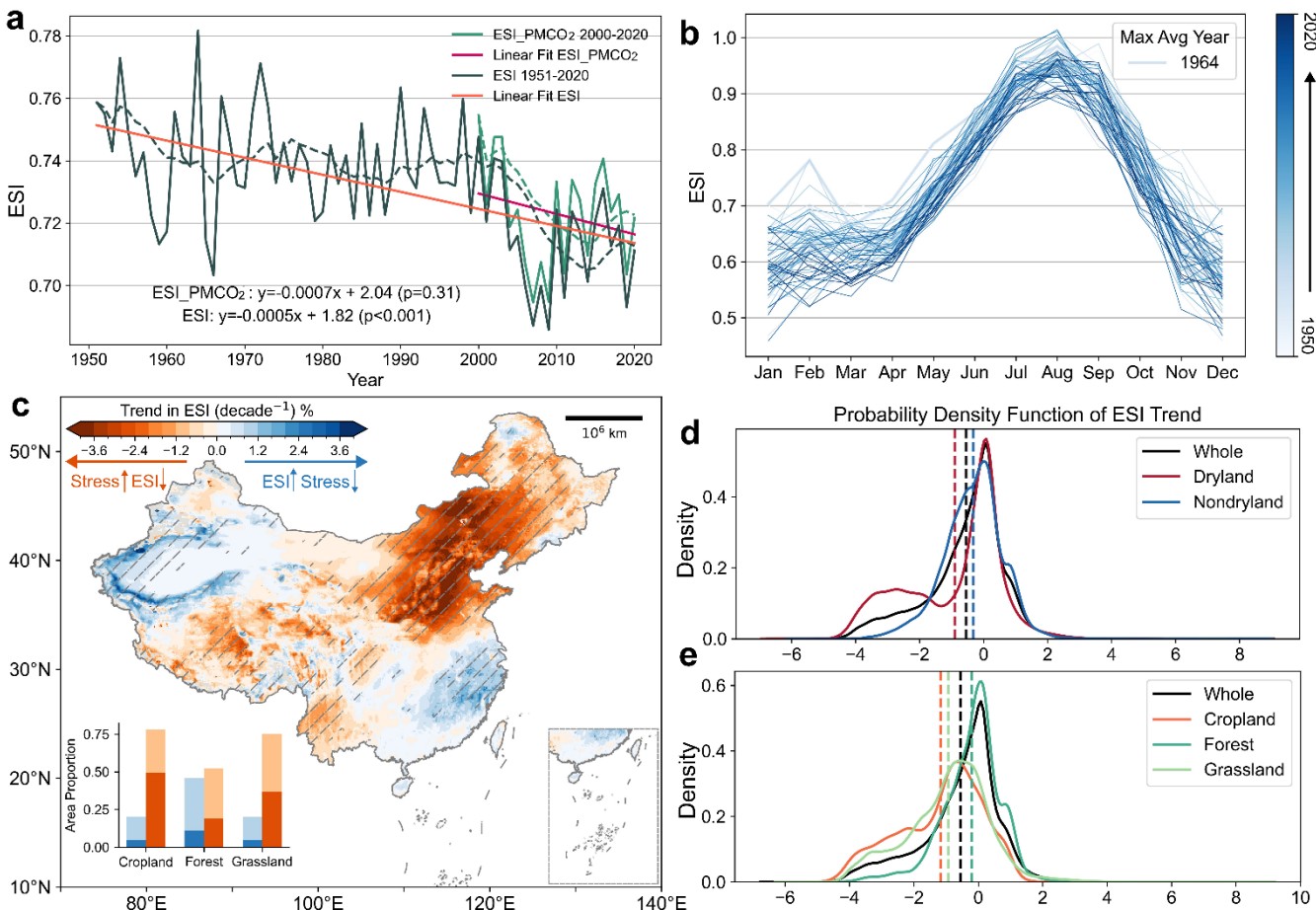

**Figure 3. Dynamic trajectory of ESI across China from 1950 to 2020.** (a) Time series and linear regression of the mean regional ESI, with the traditional PM equation-derived ESI shown in dark green and the $CO_2$ water conservation-adjusted ESI in light green. (b) Intra-annual distribution of ESI, with each line corresponding to a different year and darker hues indicating more recent years. (c) ESI trend over the preceding seven decades, quantified by Sen's slope, where lighter shaded regions denote statistically significant trends at the 0.05 level. The inset graphically summarizes the significant (darker shades) and non-significant (lighter shades) changes of ESI across various vegetation types. Panel (d) and (e) illustrate the distribution of ESI trends in both arid and humid zones, as well as across diverse land-use categories.

### 3.2 Soil moisture as the primary external driver

We assessed the relative importance of water supply, climatic, energy, and vegetation factors on the ESI through an interpretable machine learning model utilizing RF regressor and XGBoost regressor. Table S2 presents the performance metrics of two models. Despite extensive efforts to prevent overfitting through parameter optimization (Method 2.2), the relatively low validation accuracy still reveals SHAP model's susceptibility to misallocating feature importance among highly correlated variables. To address this, we generated multiple training subsets through categorical divisions of temporal scales (8-day, 16-day, monthly) and spatial partitions (drought gradient and land cover types: cropland, forest, grassland).

This approach enabled us to obtain diverse importance rankings across China's regions and derive statistically robust importance hierarchies through distribution analysis (Fig. 4, S3).

Both models pinpoint shallow soil moisture (0-7cm) as the paramount driver, indicating that its scarcity is most likely to trigger evaporative stress, especially in water-limited regions, consistent with Liebig's law of the minimum (Danger et al., 2008; Tang and Riley, 2021). Among all the vegetation proxies, LAI performs the best, likely attributable to its closest correlation with stomatal conductance (Fig. 4b). Similar importance ranking distributions were observed using RF regressor (Fig. S3b). Comparatively, the XGBoost regressor exhibited enhanced stability across multiple subsets, with minimal divergence between training and validation performance metrics (Table S2), prompting its selection for subsequent importance quantification analysis. The cross-validation reveals that, across an 8-day interval, the impact of factors associated with water, energy, climate, and vegetation on ESI variability declines sequentially, contributing 42%, 24%, 21%, and 11% to ESI, respectively (Fig. 5b). Within each category, the drivers exerting the greatest influence are svm0-7 (0.19), Rn (0.18), VPD (0.14), and LAI (0.03) (Fig. 5a). Monthly-scale analysis reveals amplified hydrological regulation (water contribution = 0.56) with concurrent suppression of vegetation and energy controls (Fig. 5b, S4, S5), demonstrating scale-dependent hierarchy in ecosystem stress controls.

Following collinear variable filtering, we identified svm0-7, Rn, T, VPD, U, P, CO2, and LAI as critical external drivers of evaporative stress (parameter set R0). Given precipitation's secondary ranking to svm0-7, we established scenario R1 substituting precipitation to examine hydrological driver divergence (Appendix C). SHAP model calibration across temporal scales revealed ESI response patterns for both variable sets. Reduced multicollinearity enhanced model stability (Table S3). ESI generally exhibits a nonlinear relationship with several critical factors (Fig. 5c-j). Notably, ESI reduces considerably with a higher VPD, T, Rn, but increases with U, LAI and svm0-7. Similar patterns are observed across analyses at various temporal resolutions, reinforcing the robustness of our findings (Fig. S4, S5). Moreover, the non-monotonic interactions between P and $CO_2$ with ESI highlight the imperative for detailed investigation into how these factors influence the exacerbation or mitigation of evaporative stress across various numerical ranges. It is essential to note, however, that apart from Rn, an escalation in temporal scale granularity generally results in a decrease in the linearity and monotonicity of the PDPs (Fig. 5c-j) for the variables. It can be ascribed to the intricate interactions introduced by data averaging, or the distortion effects of noise, which tend to obscure the relationship (Fig. 5, S4, S5).

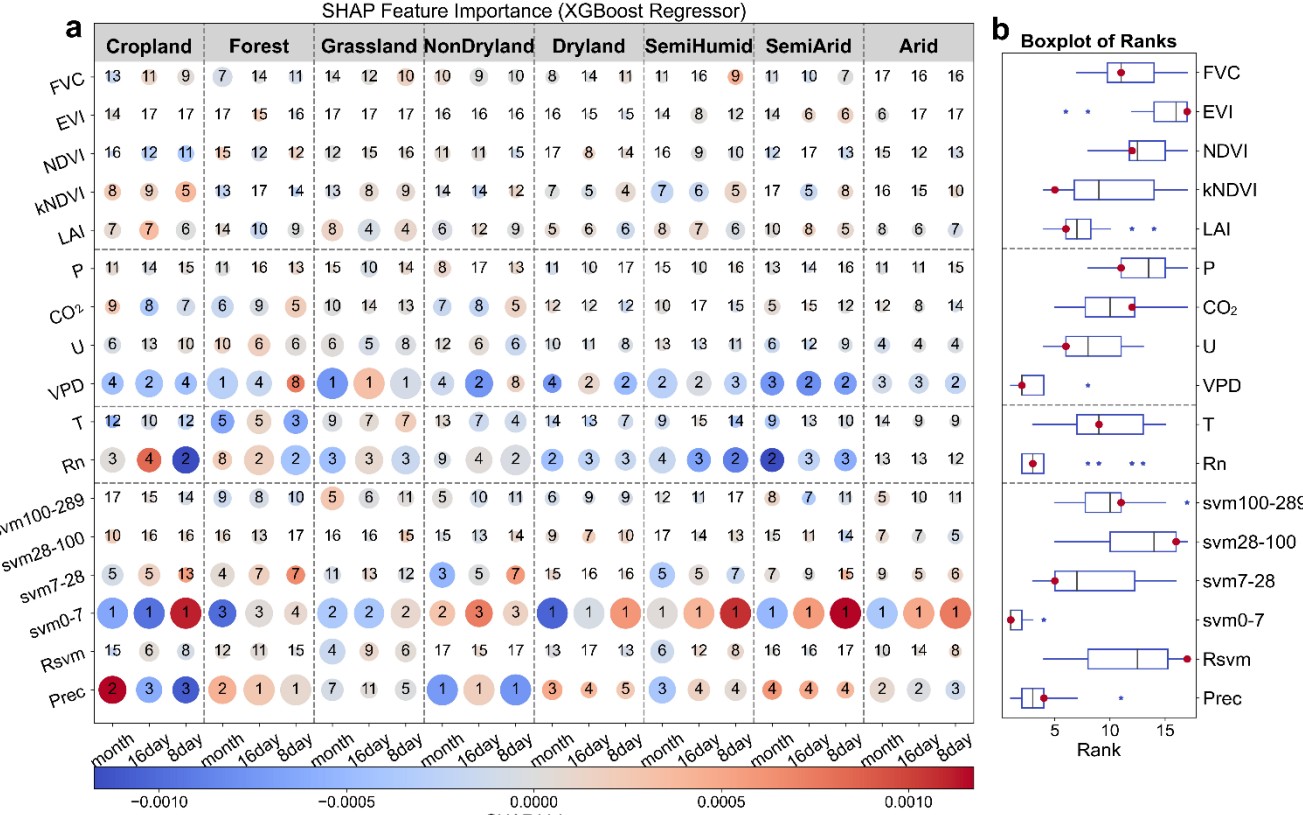

**Figure 4. Identification of the key external drivers of evapotranspiration stress index (ESI).** (a) Matrix plot employs bubble size to represent the absolute values of SHAP, with a color gradient from blue to red denoting negative and positive effects, respectively. Numerical annotations within each bubble denote the rank of importance for the variables, with higher values signifying greater influence. These ranks are obtained from an explainable machine learning utilizing XGBoost regressors. (b) Boxplot provides a statistical representation of the variable rankings across varying underlying surface conditions (corresponding to the row it's in), where the black line denotes the mean ranking and the red dot signifies the median ranking. The analysis incorporates vegetation factors such as FVC, EVI, NDVI, kNDVI, and LAI; meteorological factors including atmospheric pressure (P), carbon dioxide ($CO_2$) concentration, wind speed (U), and VPD; energy factors encompass air temperature (T) and net radiation (Rn); and water supply factors, which account for precipitation (Prec) and the soil volumetric water content (svm) across various soil layer depths, with subscripts specifying the depth range, with 0-7 indicating the soil layer from 0 to 7cm, and Rsvm representing root-zone soil moisture. Inputs to the model are the anomalies of all aforementioned variables.

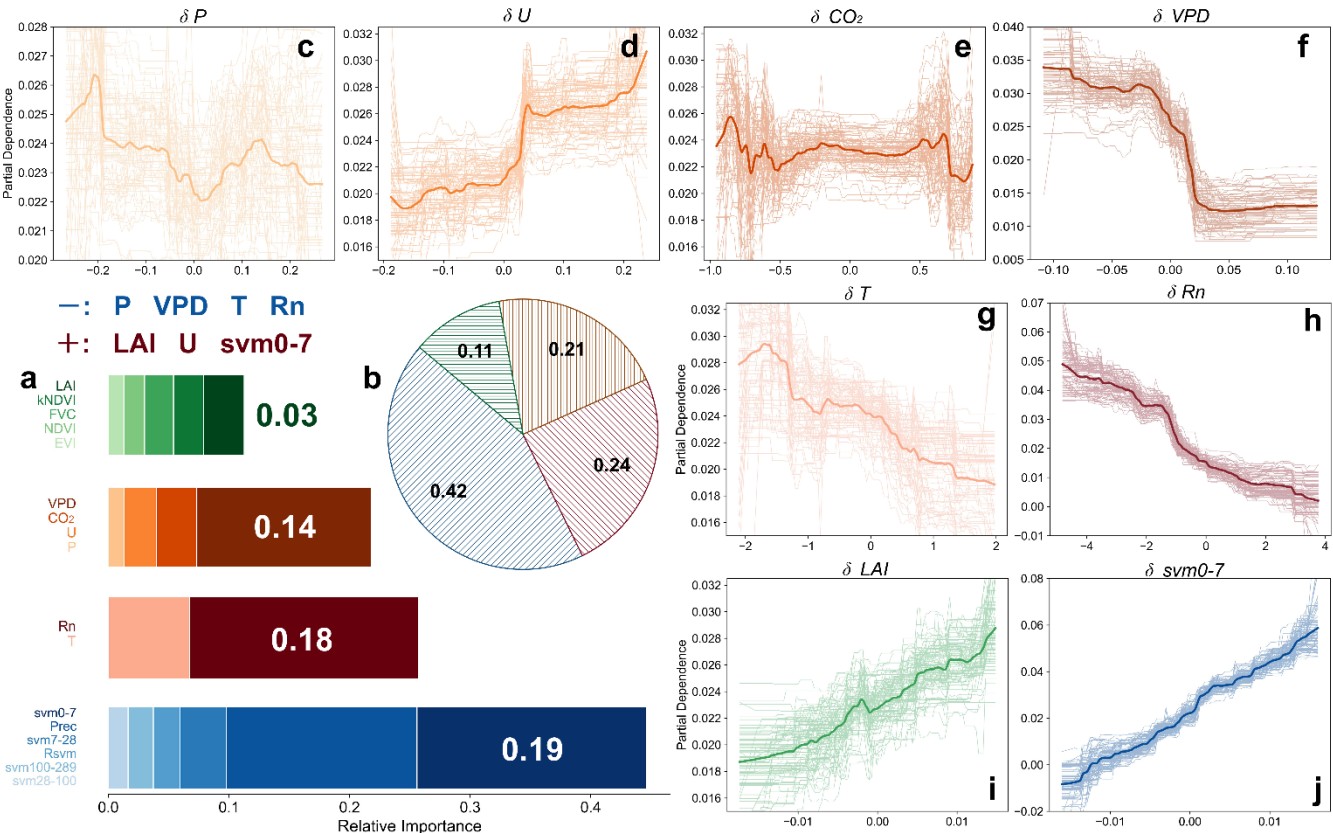

**Figure 5. Importance and partial dependence of external driving factors influencing ESI.** (a) Relative importance of hydrologic, energy, climatic, and vegetation factors in explaining ESI, with each category encompassing multiple indicators. The significance of these indicators is visually encoded by the color scheme of the bars matching their labels on the left. (b) Pie chart illustrates the proportion of the four categories described in (a). (c-j) Partial dependence graphs correspond to the following variables, in order: P, U, $CO_2$ concentration, VPD, T, Rn, LAI, and soil moisture content for the top 0-7 cm. The thickened lines indicate the average effects, 330 and the light lines around them indicate 30 random incidents from the data sets. This figure is based on an 8-day data series and generated through a SHAP model employing a XGBoost regressors, which processes the anomalies of the variables as inputs.

## 3.3 Undergoing dramatic changes in sensitivity of key drivers

To quantify the influence of these variables on the ESI, denoted by sensitivity parameter $\theta$ (Table S5), factorial simulations are conducted using the MDLM and MLR model. The findings demonstrate solid stability and powerful interpretability after 335 successfully passing the multicollinearity test (Table S4). Compared to the traditional MLR model, the estimation accuracy of the MDLM is significantly improved: the R² increases from 0.61 to 0.88, while the RMSE and MAE decrease by approximately 0.01 (Fig. 6a). Additionally, the MDLM demonstrates strong spatial consistency across the entire domain (Fig. S7~S8). Although the $\theta$ sequences for all variables estimated by both models are generally similar, differences emerge in the specific morphological fitting, particularly for $\theta_U$ and $\theta_P$ (Fig. 6gh). Notably, significant differences (p < 0.05) are observed 340 in the sequences of key factors ($\theta_{svm}$, $\theta_{VPD}$, $\theta_{LAI}$, $\theta_{Rn}$) estimated by the two models, highlighting the MDLM's enhanced

capability in estimating these sensitivities. Specifically, the MLR model generally underestimates the sensitivity of most variables (except for T and $CO_2$), exhibiting a concentrated distribution with weak sequence oscillations. However, when svm0-7 is replaced with Prec, the numerical distribution and oscillation characteristics of the parameters of the two models show an inverse relationship, which may be attributed to the interaction effects among the variable groups (Fig. 6, S6). The most pronounced linear changes in sensitivity are observed for moisture (svm0-7 or Prec), VPD, and LAI across both parameter sets (Fig. 6, S6), which are the primary focus of this study. In contrast, the contributions of other variables to sensitivity exhibit significant fluctuations and unclear trends, necessitating further investigation into their mechanisms and sources of uncertainty in future research.

Soil moisture stands out as the paramount and most sensitive factor, with its sensitivity demonstrating a significant upward trend in both models, albeit with quantitative differences (Fig. 6b). According to the MDLM results, $\theta_{svm}$ increased by 9.49% over the past two decades (0.03±0.01 yr$^{-1}$, p < 0.001), peaking in 2016, which represents a 13% rise compared to the stable level of the previous decade. In contrast, the MLR model, while producing a slightly lower mean $\theta_{svm}$ (2.36 vs. 2.98), exhibited a more pronounced growth trend (0.05±0.01 yr$^{-1}$, p < 0.001), with the peak occurring earlier in 2014. This shift in peak timing is an inherent consequence of the moving average process. As observed in similar studies, Tang et al. (2025) reported analogous evolutionary patterns and attributed this phenomenon primarily to climatic drivers. Among other pivotal drivers, $\theta_{VPD}$ and $\theta_{LAI}$ exhibit considerable numerical values and trends ( Fig. 6, S6). On average, a 1 kPa rise in VPD correlates with an estimated approximate 16.84% rise in evaporative stress based on the MDLM, which is slightly lower than the 19.74% increase estimated by the MLR. Moreover, the MLR predicts a higher annual growth rate (0.37 ± 0.07% yr$^{-1}$ vs 0.24 ± 0.05% yr$^{-1}$, p < 0.001) (Fig. 6c). Similarly, regarding the role of LAI, the MLR projects a more substantial downward trend, with a cumulative reduction of 82.16% (compared to 45.77% in the MDLM) and an annual decline rate of 0.19 ± 0.02% yr$^{-1}$ (as opposed to 0.16 ± 0.03% yr$^{-1}$ in the MDLM, p < 0.05) (Fig. 6d). This suggests that the perceived decline in the efficacy of vegetation greening for drought relief, as evaluated through conventional MLR methods, might be overstated. Moreover, the consensus result obtained using Parameter Set R1 corroborates the robustness of this observation (Fig. S6d).

Spatially, $\theta_{svm}$ is predominantly higher in the southeast and lower in the northwest, whereas $\theta_{Prec}$ exhibits a complementary pattern (Fig. 7a, S10a). This indicates that in humid regions, the ESI is more significantly affected by soil moisture regulation, while in arid regions, short-term precipitation can substantially alleviate evaporative stress. The specific mechanisms are discussed in detail in section 4.2. Additionally, $\theta_{svm}$ demonstrates considerable variation among different land-use types, with forest shows the strongest sensitivity and the most drastic changes, followed by cropland and grassland (MLDM: 4.27>3.60>1.17, Fig. 8ad). A plausible explanation is that regions endowed with optimal water conditions and robust ecological structure exhibit increased sensitivity due to the magnified impacts resulting from disturbances to the equilibrium of hydrology, climate, and vegetation (Forzieri et al., 2022). Notably, these highly sensitive hotpots distributed near the "Hu Line" and in the lower Yangtze River basin (Fig. 7a). Given the recent shifts in forest composition (large-scale planting of single tree species) coupled with the increasing frequency of extreme climatic events in these areas (Ruan et al.,

2022; Yin et al., 2022), special attention is needed. Furthermore, sizeable areas (MDLM: $4.00 \times 10^6$ km², approximately 48.3%

of China mainland) experienced a rise of $\theta_{svm}$ (Sen's Slope >0, Fig. 7a), with a similar spatial distribution also observed in MLR (Fig. S9a). However, MLR underestimates $\theta_{svm}$ in nearly all vegetated areas (~93.6%), while markedly overestimating its increasing trend across approximately 58.5% of these areas, particularly in southern regions (Fig. 7d, S9a).

Similarly, MLR also exhibits a comparable spatial pattern of underestimation in values and overestimation in trends when estimating $\theta_{VPD}$ (Fig. 7be, S9b). It exhibits notable regional disparities, with dryland encountering a 1.64-fold greater

increase in ET stress per unit of VPD compared to wet regions (MDLM: -23.02% vs -14.07%). Moreover, $\theta_{VPD}$ in arid regions is decreasing (-0.89% decade$^{-1}$, p < 0.01), in contrast to an increasing trend observed in humid regions (3.36% decade$^{-1}$, p < 0.001), so that the discrepancy between dryland and nondryland is anticipated to widen (Fig. 8be). Notably, MLR indicates an even more pronounced divergence in trends between dryland and nondryland regions (-2.41% vs. 6.58% decade$^{-1}$, p < 0.001, Fig. 8e). Interestingly, the response of evaporative stress to LAI ($\theta_{LAI}$) exhibits in both positive and

negative ways. Approximately 29.6% of the national territory ($2.81 \times 10^6$ km²), counterintuitively, experienced an increase in LAI coupled with intensified ET stress ($\theta_{LAI} < 0$), predominantly in the Loess Plateau, North China Plain, and lower Yangtze River region ( Fig. 7c). In the Loess Plateau and North China Plain, extensive research has linked rapid greening and the expansion of intensive agriculture to heightened water deficits (Feng et al., 2016; Lu et al., 2018; Wang et al., 2024). In the lower Yangtze River region, observational evidence suggests that elevated LAI reduces evaporation (Zhang et al., 2021c).

This phenomenon may arise from the shading effect of increased LAI in energy-limited areas, which diminishes radiation reaching the surface, lowers soil temperature, and thereby reduces soil evaporation (Forzieri et al., 2020). Regions where increased LAI led to alleviated stress are largely situated in proximity to the "Hu Line" ($\theta_{LAI} > 0$, $4.01 \times 10^6$ km², ~42.27%). Here, the trend of sensitivity also shows a significant decrease (Sen's slope<0, Fig. 7c). This region is a focal point for China's afforestation efforts (Zhan et al., 2023), implying that the benefits of sustained greening in reducing ET stress might

be lessening. Additionally, the regulation of artificial agronomic measures render minimal and stable $\theta_{LAI}$ in cropland (MLDM: $\theta_{LAI}$ = 1.11%, Sen's slope = -0.48% decade$^{-1}$, Fig. 8cf). Figure 8 further illustrates that the values and trends of $\theta_{svm}$, $\theta_{VPD}$, and $\theta_{LAI}$, as estimated by MLDM and MLR, display statistically significant differences (p < 0.001) across nearly all climate zones and vegetation types, underscoring substantial methodological discrepancies. Within the same land use categories, the MLR model consistently demonstrates more dispersed sensitivity trend distributions and greater intra-group

variability (Fig. 8).

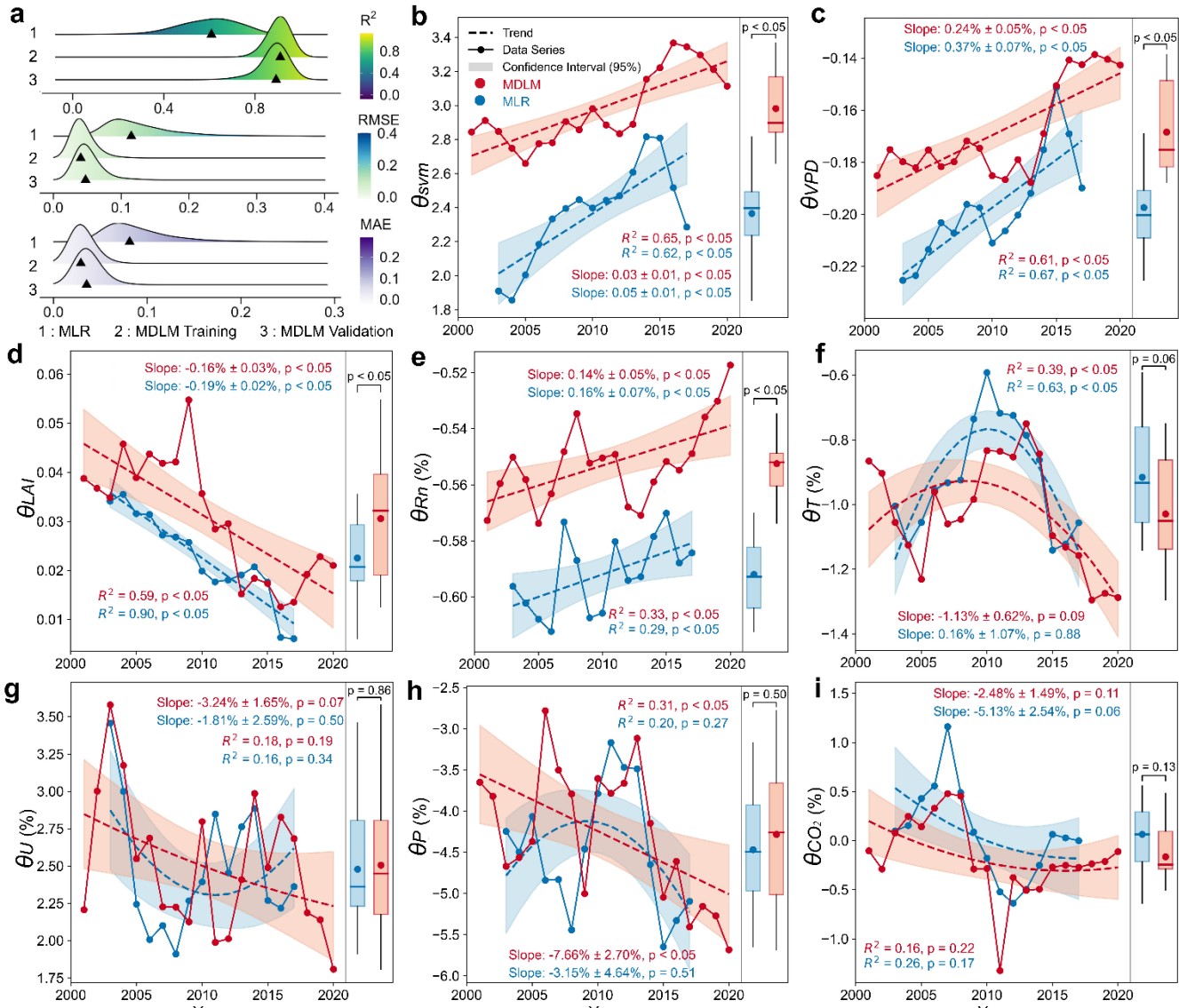

**Figure 6. Sensitivity of the ESI to external drivers and its dynamic trajectory.** (a) Distributions of R², RMSE, and MAE for MLR and MLDM models across all grid cells in training and validation sets, with triangles marking median positions. (b–i) $\theta$ time series derived from MLDM (red) and MLR (blue). Dashed lines indicate linear or quadratic fit trends, with shaded regions representing 95% confidence intervals of the fits. Text annotations display Sen's slope estimates with statistical significance, along with R² values and fit significance; quadratic fits were applied when linear trends were insignificant. Boxplots illustrate $\theta$ distributions: boxes denote interquartile ranges (25th–75th percentiles), horizontal lines represent medians, dots indicate means, whiskers span 5th–95th percentiles, and text annotations specify t-test significance of differences between model results.

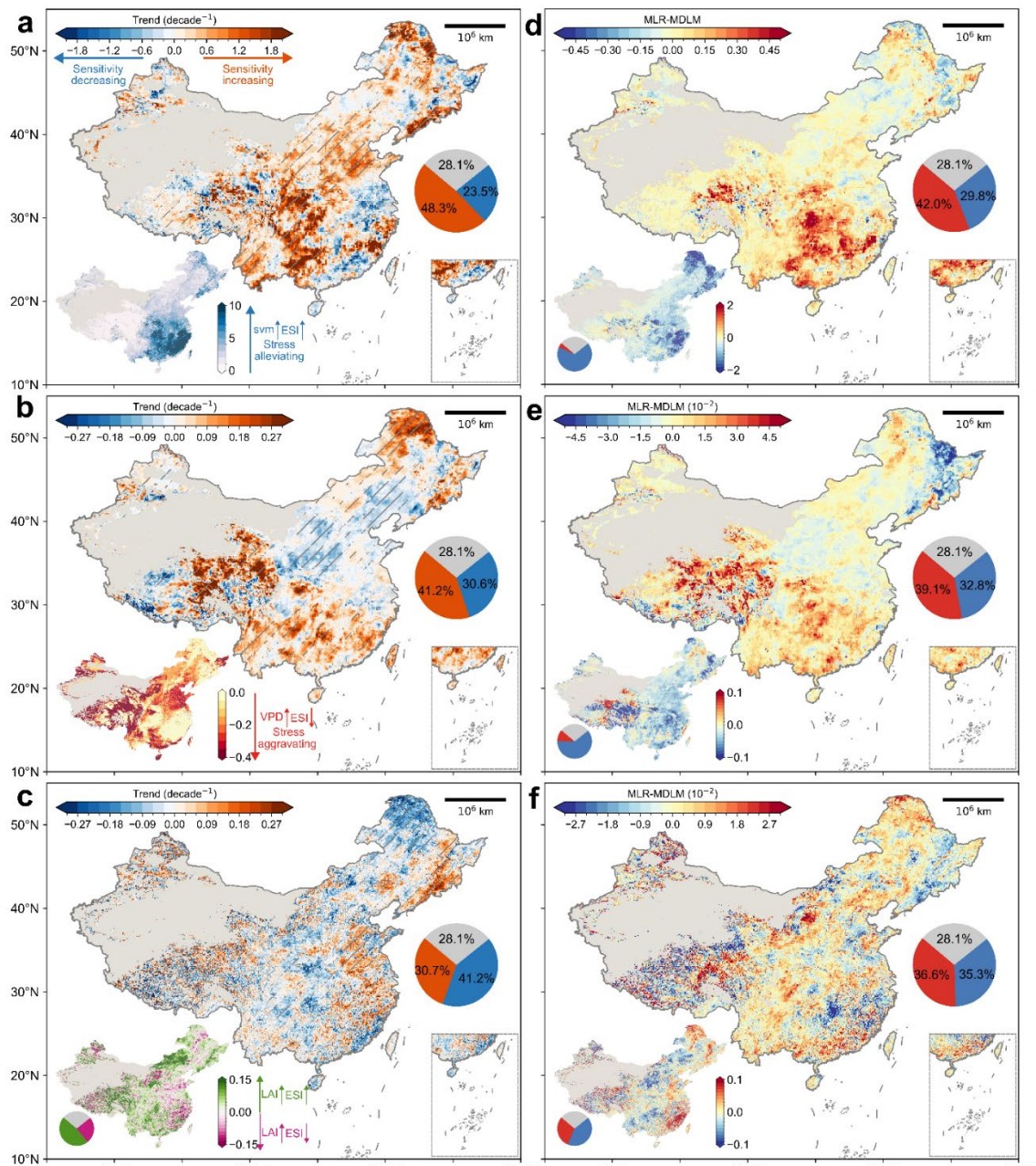

**Figure 7. Spatiotemporal heterogeneity in sensitivity of ESI to svm0-7, VPD, and LAI.** (a–c) Spatial distributions of temporal trends (estimated by Sen's slope) for $\theta_{svm}$ (sensitivity of ESI to svm0-7), $\theta_{VPD}$, and $\theta_{LAI}$, respectively. Orange indicates increasing trends, blue denotes decreasing trends, with hatched patterns marking significance at $p < 0.05$ (Mann-Kendall test). Insets display spatial distributions of multi-year mean $\theta$.(d–f) Discrepancies in Sen's slope estimates and multi-year mean values between MLR- and MDLM-derived $\theta$ (a–c). Pie charts quantify the proportion of land area exhibiting corresponding trend patterns. Gray indicates non-vegetated areas.

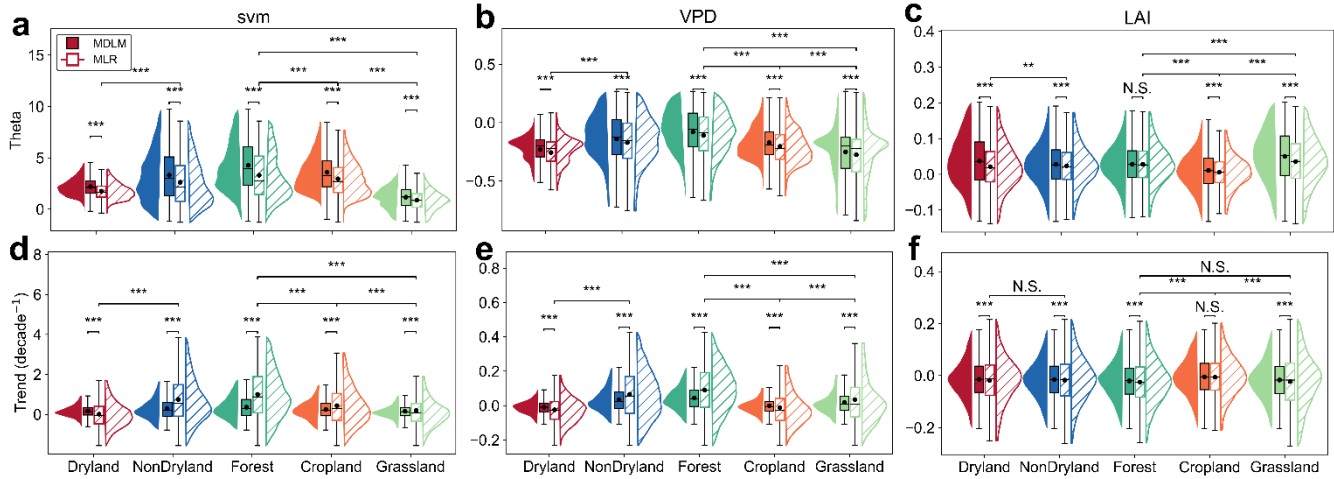

**Figure 8. Sensitivity comparisons between MDLM and DLM models across regional land types.** (a-c) Distributions of sensitivity parameters $\theta$ (ESI to svm, VPD, and LAI) in dryland, nondryland, forest, cropland, and grassland. Boxplots show interquartile ranges (25th–75th percentiles, boxes), medians (central lines), means (dots), and data ranges (5th–95th percentiles, whiskers). Corresponding probability density curves are plotted alongside. Top asterisks indicate t-test significance between different groups: three asterisks (*) denote p < 0.001, N.S. = non-significant differences.

### 3.4 Sensitivity of soil moisture increases with enhanced greening

In light of the prominent greening tendency in China, we further explored the connection between greening and the sensitivity of variables that change dramatically (Fig. 9). In extremely humid regions (AI>1.5), areas exhibiting more rapid greening trends are associated with lower $\theta_{svm}$ values (Fig. 9a). Conversely, in less humid regions (AI < 1.5), particularly in semi-arid and semi-humid areas (0.2 < AI < 0.65), there is a consistent increase in both $\theta_{svm}$ and its trend alongside the gradient of greening trends, suggesting higher sensitivity and greater acceleration (Fig. 9ab). Specifically, when 0.2 <AI<0.5, $\theta_{svm}$ exhibits a progressive increase from 1.30 to 2.20, while the associated trend simultaneously escalates from approximately 0 to 0.2. In contrast, $\theta_{VPD}$ and $\theta_{LAI}$ and their trends do not exhibit a co-varying gradient pattern along the greening trends (Fig. 9d-i). Remarkably, drier regions with slower pace of greening display greater absolute values of $\theta_{VPD}$ ($|\theta_{VPD}|$, Fig. 9d), suggesting that heightened VPD amplifies the impact of ET stress on a per-unit basis. However, in areas with rapid greening, $|\theta_{VPD}|$ tends to be diminished (Fig. 9d), as the enhanced transpiration associated with denser vegetation could elevate atmospheric moisture levels, potentially offsetting the stress from increased atmospheric moisture deficits(Fig. 9d,f). These findings imply that vegetation greening may, to some extent, enhance the ecosystem's resilience to atmospheric drought. Concerning $\theta_{LAI}$, a higher rate of increase in LAI corresponds to a more rapid decline in $\theta_{LAI}$, resulting in a reduced efficacy of LAI in mitigating evapotranspiration pressure (Fig. 9h, i). Consequently, the alleviation of water stress achieved through greening initiatives might not be as substantial as anticipated.

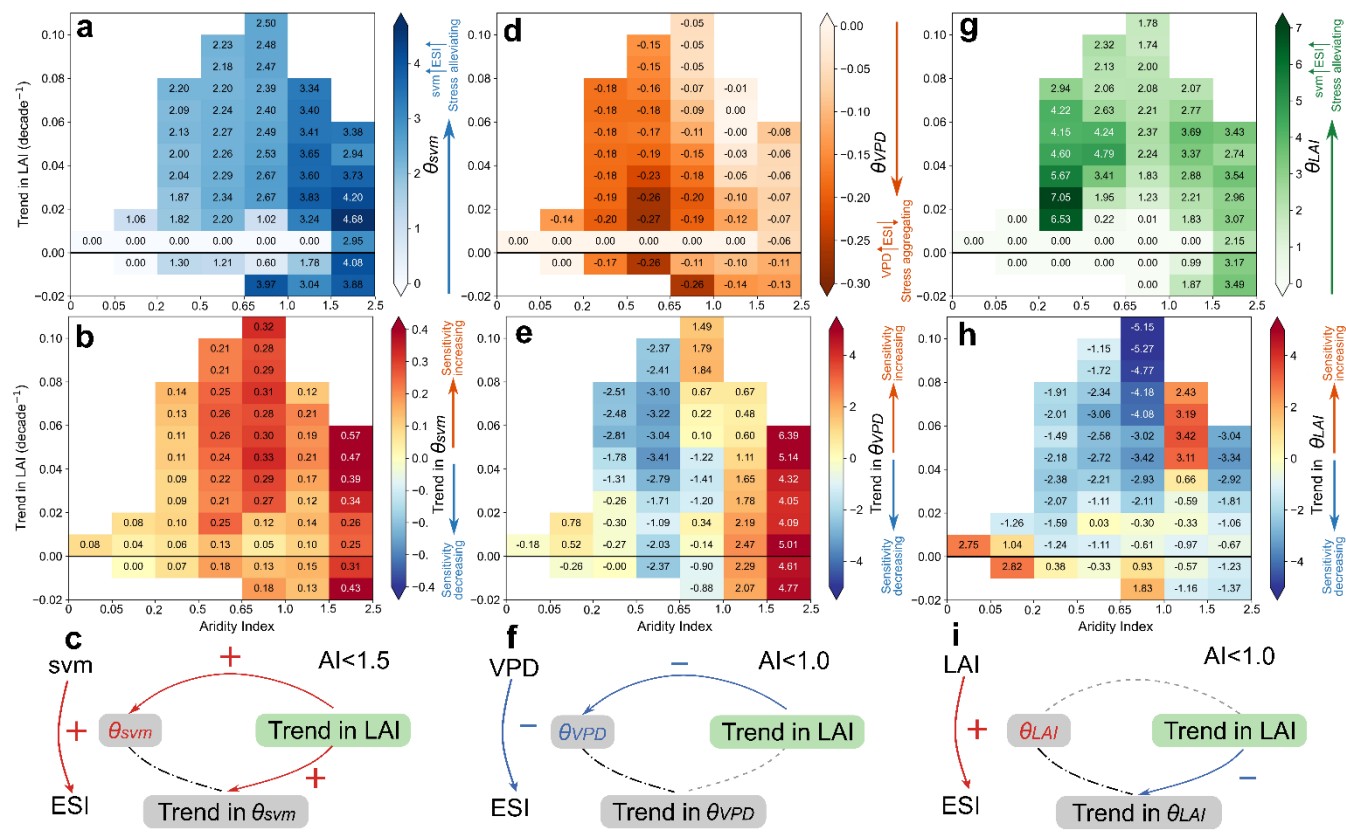

**Figure 9. Sensitivity and its trend of ESI to svm0-7, VPD, and LAI along greening and aridity gradient.** (a) The grouping statistics of $\boldsymbol{\theta}_{svm}$ across different bins, categorized by differing degrees of aridity and trends in LAI. Median value is shown for each grid and the number of grid points within each bin is shown in Supplementary Fig. S12. Similar to (a), (b) pertains to the grouping statistics of the trends of $\boldsymbol{\theta}_{svm}$. (c) The conceptual diagram delineates the covariation between LAI trends and $\boldsymbol{\theta}_{svm}$, including their respective trends, in regions with an Aridity Index (AI) below 1.5. It offers a conceptual interpretation of the patterns depicted in (a) and (b) for these areas. Note that an increase in ESI indicates a mitigation of evaporative stress. Figures (d-f) and (g-h) are analogous to (a-c); however, their protagonist is VPD and LAI, respectively. In figures (c-i), the upper right corner indicates the aridity of regions adhering to the specified pattern. A red line with a "+" sign denotes direct proportionality, whereas a "-" sign accompanied by a blue line indicates inverse proportionality. A grey dashed line represents an ambiguous relationship, and a black dashed line highlights the sensitivity parameter θ and its corresponding trend.

## 4 Discussion

### 4.1 Mechanistic Advancements of the MDLM model

Previous research has largely focused on the long-term average response of individual variables in the hydrological-climatic-vegetation nexus. Our study employs the Evapotranspiration Stress Index (ESI) as a metric to assess the dynamic equilibrium of the system and introduce the MDLM model to capture its time-varying response (Fig. 1). The MDLM model, characterized by its mechanism-driven architecture and operational simplicity, can inherently detrend and deseasonalize data,

thus eliminating the need for anomaly pre-processing. It demonstrates significant performance advantages: simulation accuracy is enhanced by 44.26% compared to moving-window MLR (with $R^2$ increasing from 0.61 to 0.88, Fig. 6a, S6~S8), consistent with high fitting accuracies (>0.90) reported in other studies (Zhang et al., 2021b). This improvement is primarily attributed to the MDLM's use of Bayesian forward filtering for dynamic parameter estimation and its capability to approximate nonlinear dependencies as quasi-linear relationships at discrete temporal nodes (Fig. 3). Consequently, it provides time-specific analyses, unlike the window-averaged sensitivity generated by MLR (Liu, 2019; Zhang et al., 2022). Pixel-scale model comparisons indicate that the MDLM model consistently exhibits superior performance across the entire study area (Fig. S7, S8). This highlights its high flexibility and broad applicability, enabling adaptation to diverse climatic conditions and terrain features. More importantly, a comparison of the $\theta$ sequences for key variables (svm, VPD, and LAI), reveals that the MLR significantly overestimates their temporal trends (Fig. 6~8, S6, S9, S10). These MLR processes— sequence truncation, smoothing, and simplification of nonlinear dynamics—not only lead to signal loss but also cause substantial differences in $\theta$ evolution trajectories (Fig. 6g, h; Fig. S6g). These findings highlight the need to re-evaluate the suitability of traditional moving average methods for determining inter-variable coupling strength or temporal sensitivity trends.

## 4.2 Enhanced Evapotranspiration Stress and Its Moisture-Driven Mechanisms

Our findings indicate that despite the water-saving benefits attributed to elevated atmospheric $CO_2$ levels, various regions across China have distinctly encountered a heightened evaporative stress (Fig. 3ab, S1). This phenomenon is in concordance with several detrimental environmental shifts recently documented, including the desiccation of land surfaces, intensification of evaporation demand, and depletion of groundwater reserves (Jasechko et al., 2024; Qing et al., 2023; Yuan et al., 2019). The pronounced increase in ET stress in dryland and cropland underscores the vulnerability of these ecosystems (Fig. 3cde, S1, S2ac). This is primarily attributed to the considerable desiccation of soil moisture (Fig. S11a), which serves as the exclusive direct water-supply source for terrestrial evaporation processes—whether through evaporation of land surface or transpiration via plant stomata—and thus exerts a paramount role (Smith and Boers, 2023; Zhao et al., 2023)( Fig. 4~6, S3~S6-6 ). In the past 20 years, due to increased soil water sensitivity, soil water variability has led to greater ET stress intensification (Fig. 6b, 7a9, S11a). Sporadic studies on time-varying sensitivity also reveal that water supply has a gradually increasing impact on ecosystems (Hu et al., 2023; Li et al., 2022a; Zeng et al., 2022; Zhang et al., 2022). Our simulation scenarios using precipitation as a proxy for moisture also corroborated these findings (Fig. S6b, S10). Hence, addressing the threats posed by soil moisture deficits to water and food security in arid cropland becomes imperative when implementing agricultural practices. Intriguingly, our research identifies a complementary spatial pattern between $\theta_{svm}$ (southern high-northern low) and $\theta_{Prec}$ across China (Fig. 7a, S10a). In northern arid/semi-arid regions characterized by low initial soil moisture, precipitation acts as an "instantaneous input" that rapidly replenishes soil moisture and substantially alleviates evaporative stress, exhibiting higher sensitivity. Conversely, in southern humid regions where soils maintain near-saturation

moisture levels, surpassing the pre-existing hydro-equilibrium threshold triggers disproportionately severe ecosystem perturbations. Consequently, ESI in these regions demonstrates heightened dependence and sensitivity to soil moisture dynamics. Analogously, despite a relatively moderate increase in forest ET stress (Fig. 3cde, S2), soil moisture sensitivity emerges both highly pronounced and rapidly enhanced in recent years (Fig. 8a). In regions with favorable hydrological conditions and stable ecological structures, the collapse of the hydrological-climatic-vegetation system can precipitate profoundly adverse effects (Forzieri et al., 2022). Furthermore, in the face of increasingly intense and severe extreme climatic events, complex ecosystems require enhanced scrutiny to formulate proactive strategies.

### 4.3 Ecological Implications of "Greening but Drying" Feedbacks in a Changing Climates

Additionally, the greening pace may alter the ability of ecosystems against multiple types of water stress. On the one hand, in semi-arid and -humid regions, the sensitivity of ESI to soil moisture and its variability are proportional to the greening trend (Fig. 9abc). Therefore, once a soil drought occurs, the adverse impact on ET stress increases, suggesting that overly rapid greening can diminish an ecosystem's resilience to soil drought. Specifically, when afforestation surpasses the carrying capacity dictated by regional hydroclimatic conditions, a slight soil deficit can trigger the self-amplify cycle and result in the degradation and mortality of vegetation, known as overshoot drought (Zhang et al., 2021b). From a long-term lens, the reliance on irrigation in the initial stages of artificial vegetation establishment, aimed at ensuring plant survival, can hamper the formation of deep root systems (Moreno-Mateos et al., 2020; Xiao et al., 2024b). Such a practice undermines the plant-groundwater linkage in dryland, subsequently diminishing their resilience (Wang et al., 2023). On the other hand, VPD sensitivity and the greening trend exhibit an inverse gradient (Fig. 9d), particularly in arid lands with high $\theta_{VPD}$ values Fig. 8b), indicating that greening in drylands has improved the ability to cope with atmospheric drought. Two mechanisms may explain this phenomenon: (1) Deep-rooted vegetation in certain arid zones accesses deep soil moisture during droughts, releasing it into the atmosphere via transpiration to directly elevate humidity—though this may simultaneously intensify soil aridity (Liang et al., 2024; Sun et al., 2021). (2) Greening lowers surface albedo, suppressing surface temperature increases and thereby mitigating atmospheric drought through indirect thermal regulation (Zhang et al., 2024). Concerning LAI, a more rapid greening trend correlates with a diminishing contribution of LAI growth to the alleviation of water stress (Fig. 9d, 8h, S13h). Furthermore, in certain regions, an enhanced LAI may exacerbate water stress (Fig. 7c). Gleason et al., (2017) has found that high vegetation density may exacerbate inter-species competition for water, leading to increased water stress for individual plants or specific plant communities, providing a cogent explanation for our outcome. Collectively, although greening has traditionally been considered a positive environmental adjustment, we advocate for a more critical and dialectical understanding. In adherence to sustainable development imperatives, policymakers and decision-makers are tasked with the precise identification of regions amenable to greening initiatives and the establishment of quantifiable benchmarks that will safeguard ecological equilibrium and foster enduring sustainability.

## 4.4 Research Limitations and Prospects

Although this study provides interesting results for the interplay between regional greening and complex ecological couplings in China, this study remains subject to certain limitations in methodology and data. Firstly, the observational sample size from flux stations is insufficient, as it has been employed solely for the screening of reanalysis products rather than for dynamic factorial simulations. Secondly, the deviations observed in Parameter Set R0 and R1 indicate that other local factors, such as soil texture, vegetation acclimation, plant demographic rates, and vegetation-groundwater dependency, may exert considerable influence (Abel et al., 2024; Fu et al., 2024b; Patel et al., 2021). More in-depth studies based on long-term ecophysiological observations and purpose-built field experiments are required to further unravel the complex mechanisms of VPD, soil moisture, and LAI in land-atmosphere interactions.

Finally, what are the key implications given the foreglimpse of intensifying evapotranspiration stress coinciding with its increasing sensitivity to soil water deficits in a greening China? Soil water variability would lead to greater changes in the ecosystems undergoing rapid greening. That is, stronger drought effects can be expected when soil water is anomalously low. Concurrently, the increase in LAI may not alleviate evaporative stress as much as expected. Combined with the increasing frequency and severity of extreme climate events, a "greening but drying" trend may thus be more prevalent in the future, potentially raising the risk of ecosystem imbalance.

## 5 Conclusions

Our study introduces a memory-driven dynamic linear model (MDLM) that integrates the "dry-get-drier" legacy effects to evaluate time-specific sensitivities of evapotranspiration stress drivers in China under rapid vegetation greening. Using the Evapotranspiration Stress Index (ESI) as a proxy for ecosystem water-atmosphere-vegetation equilibrium, we documented a 4.74% intensification of ET stress across mainland China from 1950 to 2020. Soil moisture sensitivity was identified as the dominant driver, showing a 9.49% increase during 2001–2020. Enhanced vegetation greening exhibited stronger coupling with elevated soil moisture sensitivity but reduced VPD sensitivity, reflecting greater susceptibility to soil drought concurrent with improved resilience to atmospheric aridity. Compared to the conventional moving-window multiple linear regression (MLR) method, the MDLM framework increased the coefficient of determination ($R^2$) by 44.26%, significantly improving sensitivity estimates for critical drivers. Methodologically, this work underscores the need to critically re-evaluate trend overestimation inherent in traditional MLR methods. Our findings advance mechanistic understanding of complex regional ecosystem dynamics and offer guidance for steering greening strategies toward a more stable equilibrium among water, atmosphere, and vegetation.

**Appendix A: Calculation and validation of the ESI**

We utilize the ratio of actual evapotranspiration (ETa) to potential evapotranspiration (ETp) to calculate the Evapotranspiration Stress Index (ESI). Prior to investigating the dynamic changes of ESI, we compared the representativeness of three mainstream datasets: the Global Land Evaporation Amsterdam Model v3.7a (daily, 0.25°), the Terra Moderate Resolution Imaging Spectroradiometer (MODIS) MOD16A2GF Version 6.1 (8-day, 500 m), and the European Center for Medium-Range Weather Forecasts (ECMWF) ERA5-Land (hourly, 0.1°). The results are subsequently validated against observational data from 26 eddy covariance (EC) stations (Fig. A1,), which are sourced from various observation networks (Information regarding EC stations is provided in Table S1 in Supporting Information). Each site has at least one year of continuous observations and energy balance residuals < 35 W m$^{-2}$ (calculated using Eq. A1). Before calculation, outlier observations are eliminated using a three-standard-deviation method, and observations recorded on rainy days are excluded. Subsequently, data imputation is conducted using the IterativeImputer tool in Python. The latent heat measured at the EC station is converted to ETa (Eq. A2), while ETp is calculated using the traditional Penman-Monteith method (Eq. A3).

$$Residual = R_n - G - LE - H \tag{A1}$$

where $R_n$ represents net radiation, $G$ denotes soil heat flux, which is zero on a daily scale, $LE$ refers to daily latent heat, and $H$ represents sensible heat flux, with all units expressed in W m$^{-2}$.

$$ET_a = \frac{LE}{\lambda} \tag{A2}$$

In Eq. A2, the unit of $ET_a$ is mm day$^{-1}$, with the constant value of $\lambda$ at 0°C taken as 28.94.

$$ET_p = \frac{0.408\Delta(R_n - G) + \gamma\dfrac{900}{T + 273}U(e_s - e_a)}{\Delta + \gamma(1 + 0.34U)} \tag{A3}$$

where the unit of ETp is mm day$^{-1}$, $\Delta$ is the slope of the saturation vapour pressure versus temperature curve (kPa °C$^{-1}$), $\gamma$ is the psychrometric constant (kPa °C$^{-1}$), $U$ is wind speed at 2 m (m s$^{-1}$), T is surface air temperature (°C), $e_s$ and $e_a$ are saturated and actual vapour pressure (kPa) and whose difference serves as a crucial indicator of atmospheric moisture deficit, known as vapor pressure deficit (VPD). At the grid scale, all climatic variables used to calculate ETp, including Rn, T, and U, are obtained from the ERA5 dataset. In this dataset, VPD is calculated based on the Clausius-Clapeyron relation using 2m temperature and dew point temperature (Held and Soden, 2006; Zhong et al., 2023).

To prevent the overestimation of ETp, we employed the formula developed by Yang et al., (2019) to calculate grid ETp while accounting for the $CO_2$ water-saving effect (Eq. A4). This formula has been widely recognized and applied in recent studies, providing an important theoretical basis for accurately estimating evapotranspiration (Lian et al., 2021; Zhang et al., 2023). All meteorological data used in the formula were sourced from ERA5, and the monthly $CO_2$ data were obtained from CarbonTracker (CT2022) for the period from 2001 to 2020, with a spatial resolution of 3° × 2°. For comparison, after

considering the water-saving effect of $CO_2$, the mean, maximum, and minimum values of ETp in China decreased by 9.48 mm, 30.85 mm, and 19.42 mm, respectively, while the pattern is mirrored (Fig. A2abd). The conventional ETp calculation method overestimated values in 99.6% of China mainland, particularly in northern arid regions (Fig. A2c).

$$ET_p = \frac{0.408\Delta(R_n - G) + \gamma\dfrac{900}{T + 273}U(e_s - e_a)}{\Delta + \gamma\{1 + U[0.34 + 2.4 \times 10^{-4}([CO_2] - 300)]\}} \tag{A4}$$

where $2.4 \times 10^{-4}([CO_2] - 300)$ denotes the effect of atmospheric $CO_2$ concentration (ppm) on surface stomatal resistance.

Figure A3 compares the ESI derived from three datasets, revealing substantial discrepancies attributable to methodological differences. For ETp estimation, MODIS and ERA5 apply the PM formula calibrated for idealized vegetation types, consistent with our research hypotheses in Figure 1. In contrast, GLEAM integrates corrections for actual vegetation heterogeneity. The ETa < ETp constraint imposed on MODIS and GLEAM limits their ability to capture water surplus conditions (Figure A3cd), whereas ERA5—free of this restriction—produces ESI spatial patterns aligned with eddy covariance (EC) flux measurements (Fig. A3ab). Incorporating $CO_2$ water-saving effects into ETp calculations identifies an expansion of non-water-limited regions, enhancing aridity-humidity characterization ontological congruence (Fig. A3abe). Note that our research is dedicated to analyzing the temporal dynamics of the ESI, particularly in relation to the intensification of drought conditions. The conceptual diagram (Fig. 1) demonstrates that lower ESI values signify increasingly severe imbalances and stress levels. The study does not address how often the threshold of 1 is crossed nor explores the physical significance of this threshold.

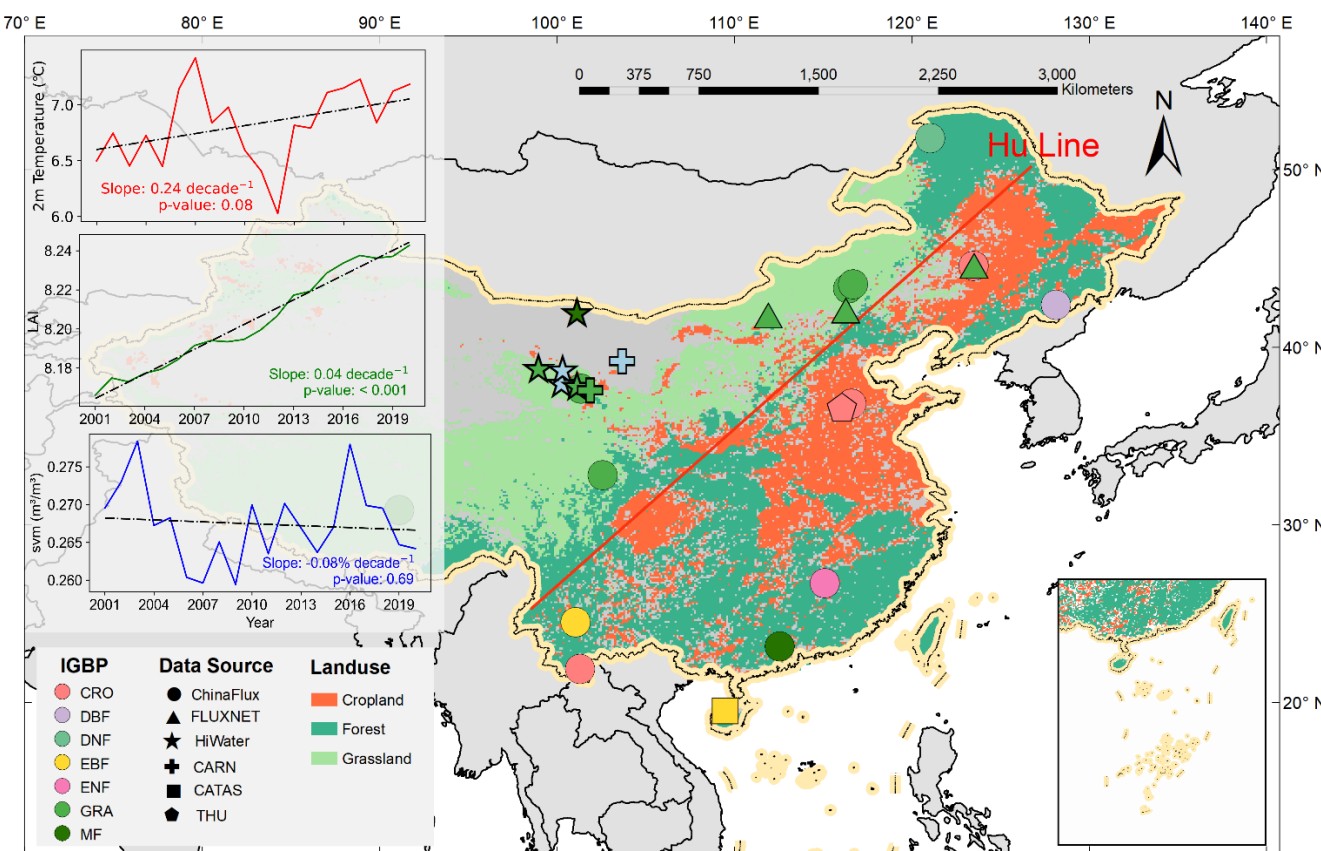

**Figure A 1. Location of the study area and distribution of multi-year average ESI.** The extent of the study area, employing a tri-chromatic map to differentiate between cropland, forest, and grassland; dots denote the positions of EC flux stations, with colors representing the International Geosphere-Biosphere Programme (IGBP) classifications and shapes indicating observational sources. The inset displays time-series plots of regional average air temperature (red), surface soil moisture (blue), and LAI (green) over the past 20 years, with a black dashed line denoting the linear trend. The red slanted line denotes the position of the "Hu Line (also known as the Hu Huanyong Line)", with its neighboring transitional area defined as the "Hu Zone",a fundamental geographical demarcation in China that characteristically differentiates regional hydrological regimes, vegetation coverage gradients, and demographic distribution patterns (Hu, 1935; Li et al., 2024).

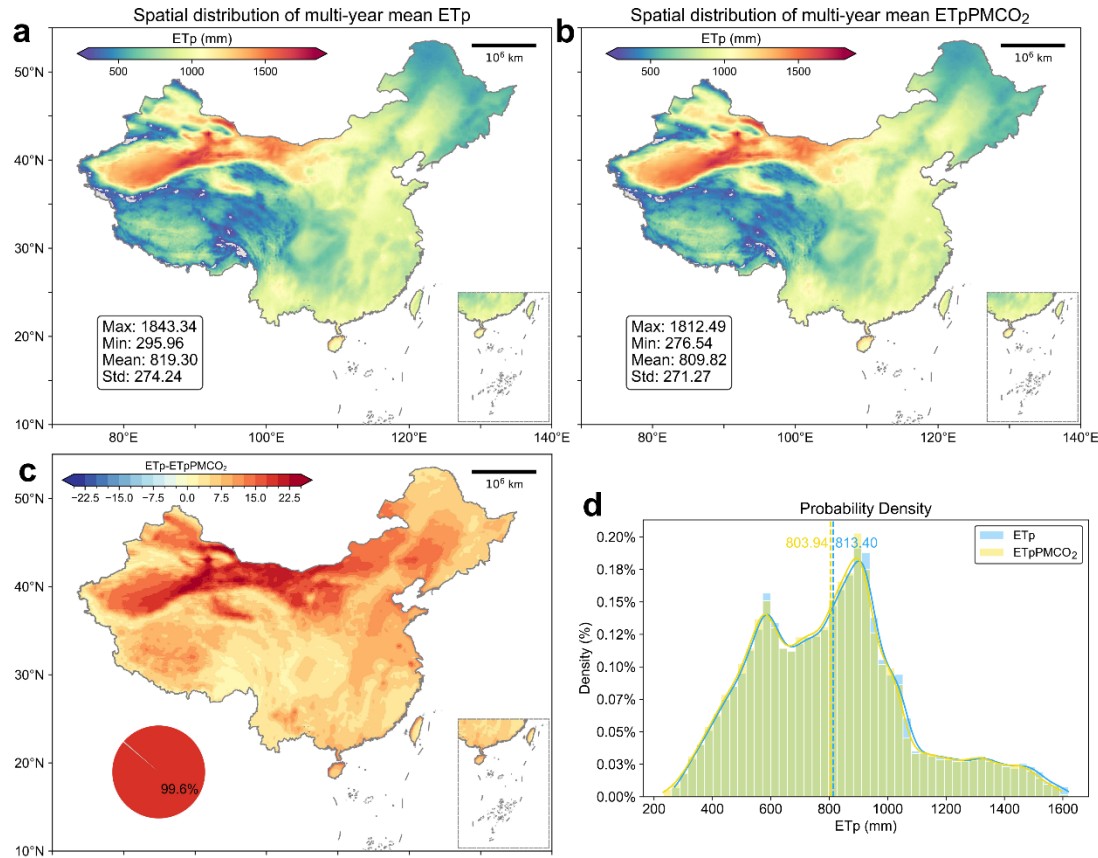

**Figure A 2. Comparison of the spatial distribution of the multi-year average potential evapotranspiration (ETp).** (a) calculated by the traditional PM equation and (b) the modified PM equation that incorporates the $CO_2$ water-saving effect. The maximum value, minimum value, and average value are marked in the lower left corner of the figure. (c) the spatial distribution of the difference between ETp and ETpPMCO$_2$. (d) the probability density curves of the two.

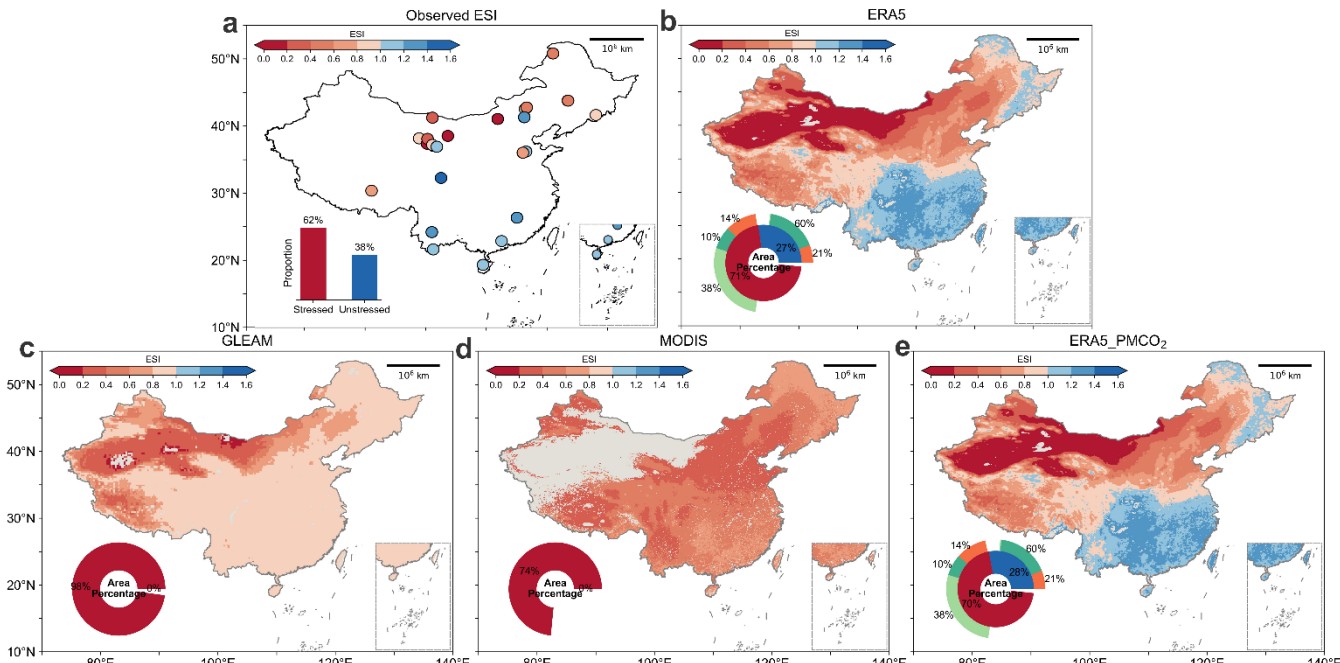

**Figure A 3. Spatial distribution of the multi-year average of ESI calculated from ETa and ETp products from different datasets.** (a) was calculated using flux tower observational data, (b) derived from ERA5, (c) originated from GLEAM, (d) obtained from MODIS, and (e) sourced from ERA5 but incorporates the CO2 water-saving effect in ETp calculation.

**Appendix B: Spatiotemporal patterns of Rsvm in China**

Previous studies frequently defined the 0-100 cm soil layer as the root zone. However, since the root zone constitutes the most dynamic interface for water-vegetation-atmosphere interactions, employing more realistic root zone soil moisture data as an indicator for evapotranspiration stress is of critical importance (Gao et al., 2014, 2024; Wang-Erlandsson et al., 2016). This research utilizes root depth data corresponding to an 80-year drought recurrence interval, estimated by Stocker et al. (2023) through the mass accumulation curve methodology. We calculated actual root zone soil moisture (Rsvm) using the weighted averaging method with layered soil moisture data from ERA5 (2001-2020). For cases where root depth exceeded 289 cm (the maximum depth covered by ERA5 data), the weighted average soil water content from 0-289 cm was adopted as the Rsvm for this pixel. The computational procedure is detailed below.

The ERA5 soil moisture dataset is stratified into four layers (Layer 1-4), representing soil strata with thicknesses of 0-7 cm, 7-28 cm, 28-100 cm, and 100-289 cm respectively. For each pixel, the actual thickness of each ERA5 layer contributing to the root zone was calculated according to root depth parameters:

Layer1 (0~7cm):

$$d_1 = min(7, Root\ Depth) \tag{B1}$$

Layer2 (7~28cm), activated only when Root Depth>7cm:

$$d_2 = max\ [0, min(28, Root\ Depth) - 7] \tag{B2}$$

Layer3 (28~100cm) , activated only when Root Depth>28cm:

$$d_3 = max\ [0, min(100, Root\ Depth) - 28] \tag{B3}$$

Layer4 (100~289cm) , activated only when Root Depth>100cm:

$$d_2 = max\ [0, min(289, Root\ Depth) - 100] \tag{B4}$$

The Rsvm is:

$$Rsvm = \frac{d_1 * svm_1 + d_2 * svm_2 + d_3 * svm_3 + d_4 * svm_4}{Root\ Depth} \tag{B5}$$

where $Root\ Depth$ denotes actual root depth (cm), $d_{1\sim4}$ represent weighted thicknesses of individual layers (cm); $svm_{1\sim4}$ indicates soil moisture values per layer,$Rsvm$ corresponds to root zone soil moisture ($m^3 \cdot m^{-3}$).

Figure B1 investigates the spatiotemporal patterns of root zone soil moisture across China. The regional mean root depth is 2.33 m, with northern arid/semi-arid regions dominated by shallow-rooted vegetation (e.g., grasslands and shrubs) exhibiting lower root zone moisture values ($<0.30\ m^3 \cdot m^{-3}$). Deeper root systems ($>15$ m) occur in northwestern arid zones and southern karst regions, where vegetation utilizes groundwater via deep-rooting adaptations (Fig. B1a). The 20-year mean Rsvm of 0.33 $m^3 \cdot m^{-3}$ displays a pronounced south-to-north gradient, correlating with precipitation gradients (Fig. B1b-d). Notably, over 50% of vegetated areas exhibit declining Rsvm trends, particularly in northwestern China and the North China Plain, where rates exceed 0.002 $m^3 \cdot m^{-3} \cdot yr^{-1}$ (Fig. B1c). These regions experience mean annual temperatures of 18–26°C and annual precipitation below 900 mm (Fig. B1d). These soil desiccation trends likely arise from climatic stressors and unsustainable groundwater extraction, requiring urgent policy interventions to mitigate threats to agricultural productivity and ecosystem stability. Conversely, northeastern and southeastern China demonstrate rising Rsvm trends, where annual precipitation remains abundant. This hydrological shift requires systematic assessment of potential reductions in flood mitigation capacity, underscoring the importance of enhanced monitoring protocols for watershed management systems to address evolving hydrological risks.

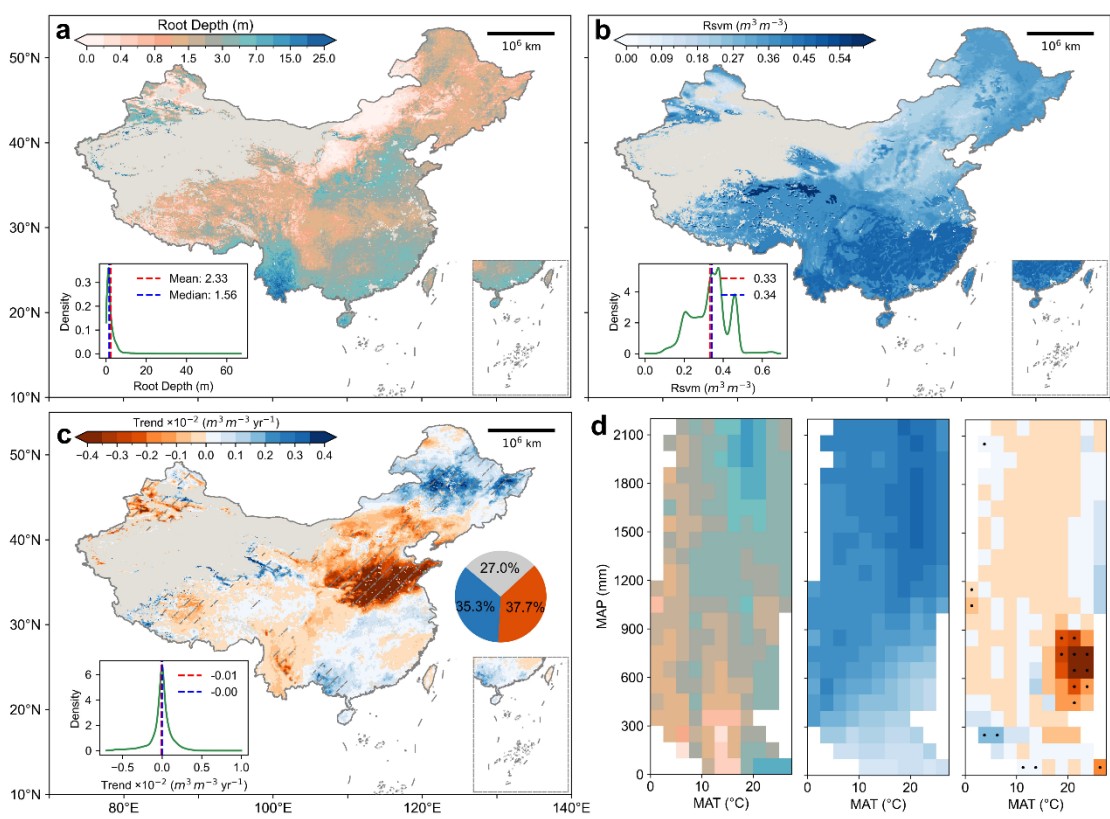

**Figure B 1. Spatiotemporal patterns of root-zone soil moisture (Rsvm) in China.** (a) Spatial distribution of root depth. (b) Multi-year mean Rsvm (2001-2020). The inset illustrates the probability density distribution with red and blue markers denoting the mean and median values, respectively. (c) Sen's slope of Rsvm during 2001-2020, where shaded areas indicate regions with statistically significant trends (p < 0.05). The inset displays areal proportions of Rsvm changes: orange represents decreasing trends, blue increasing trends, and gray indicates no-data regions. (d) Climatological phase space diagram corresponding to (a)-(c), with the x-axis showing multi-year mean air temperature and the y-axis representing precipitation. Color gradations within bins reflect median values, while dots mark regions exhibiting statistically significant Rsvm trends (p < 0.05).

## Appendix C: Comparative impacts of precipitation and surface soil moisture on ESI

Given that precipitation exhibited the second-highest statistical rank among parameters (surpassed only by svm0-7), we developed a comparative Parameter Set R1 using precipitation anomalies to investigate the distinct influences of precipitation and soil moisture on the ESI. SHAP model analysis across multiple temporal scales in China revealed contrasting feature importance patterns between parameter sets (see Table S3 for model performance metrics): R1 displayed broader contribution distributions indicating balanced parameter interactions, while R0 prioritized dominant parameters (svm0-7, Rn). Both configurations consistently showed Rn, VPD, and T negatively regulating ESI, contrasting with positive regulation by LAI and U (Fig. C1). Notably, Prec and svm0-7 synergistically enhanced ESI through moisture supply mechanisms (Fig. C1, C2, 5j, S4j-S5j), consistently ranking as top predictors. In R1, Prec showed persistent secondary influence to VPD, whereas SVM0-7 dominated feature importance at both 16-day and monthly scales (Fig. C1b-c).

Diverging from soil moisture's linear relationship, precipitation anomalies induced a biphasic "plateau-ramp" ESI response, requiring anomalies exceeding -5 to 0mm for stress mitigation. Conversely, svm0-7 exhibited acute sensitivity with near-vertical monthly response gradients (Fig. S5j), indicating immediate drought relief from minimal moisture replenishment (-0.02 to -0.01 $m^3 \cdot m^{-3}$) under extreme deficits. Our deseasoned-detrended anomaly analysis inherently accommodates negative values. Subsequent research should prioritize identifying precise response thresholds and turning points.

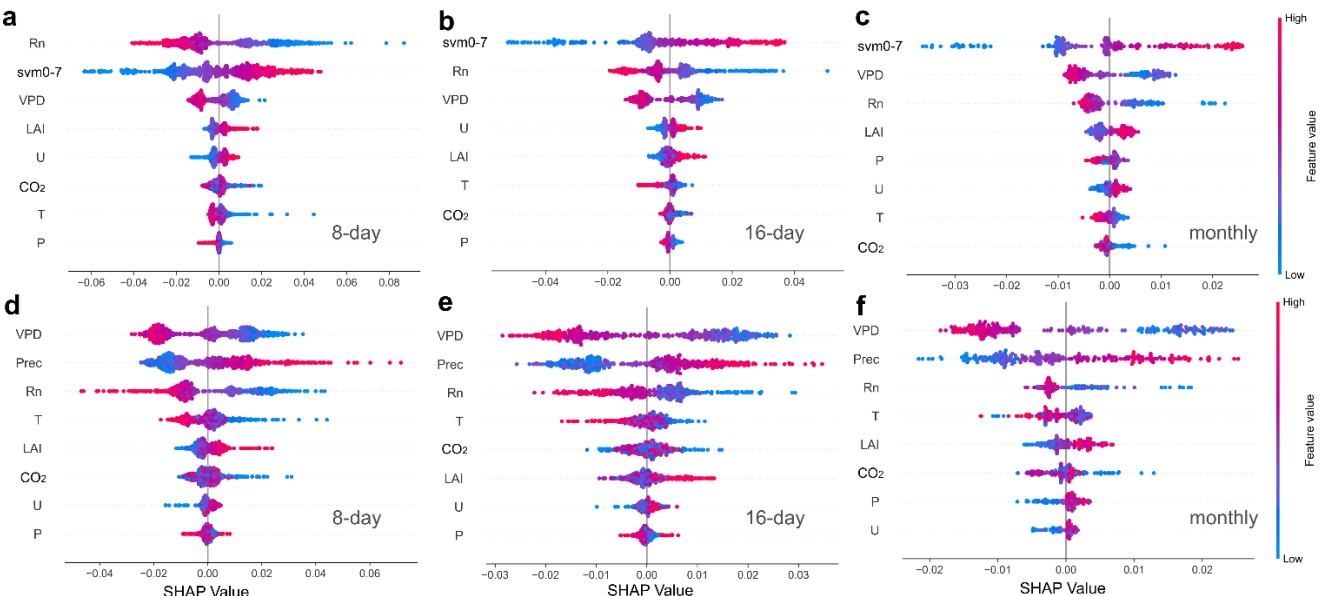

**Figure C 1.** Multi-temporal beeswarm plots between Parameter Set R0 (a-c) and R1 (d-f). Parameter set R0 includes: svm0-7, Rn, T, VPD, U, P, $CO_2$, LAI, with R1 substituting svm0-7 with Prec. Red denotes high feature values with substantial impacts on ESI. High feature values distributed along the SHAP positive half-axis signify positive contributions to ESI. All features are arranged in descending order of contribution magnitude, where wider x-axis distributions reflect stronger model impacts.

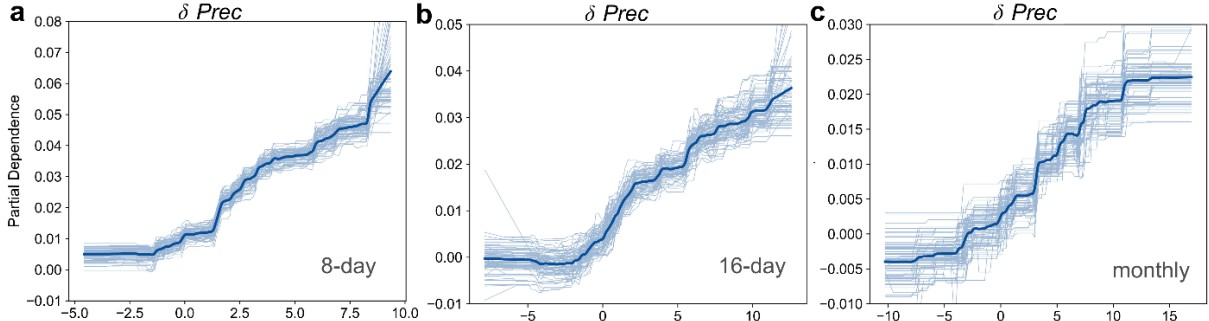

**Figure C 2.** Precipitation PDPs under Parameter Set R1 across temporal scales, with soil moisture counterparts shown in Figs. 5j, S4j, and S5j.

*Data Availability Statement.* Detailed sources of observations from 26 Eddy Covariance Flux Towers are shown in Table S1 in Supporting Information. The Global Land Evaporation Amsterdam Model v3.7a can be downloaded from the GLEAM
home/landing page (https://www.gleam.eu/). The Terra Moderate Resolution Imaging Spectroradiometer (MODIS) MOD16A2GF Version 6.1 is accessed from https://lpdaac.usgs.gov/products/mod16a2gfv061/. Meteorological variables are available from The European Center for Medium-Range Weather Forecasts (ECMWF) ERA5-Land (https://cds.climate.copernicus.eu/cdsapp#!/dataset/reanalysis-era5-land?tab=overview). The monthly $CO_2$ data were obtained from CarbonTracker (CT2022) (https://gml.noaa.gov/ccgg/carbontracker/download.php). LAI (V60) and FVC
(V40) from the Global Land Surface Satellite (GLASS) datasets are obtained from http://www.glass.umd.edu/Download.html. NDVI and EVI in version MOD13C1 are derived from https://lpdaac.usgs.gov/products/mod13c1v061/. The annual China Land Cover Dataset with a resolution of 30 meters originates from https://zenodo.org/records/8176941.

*Author contributions.* Conceptualization: YL; Data curation: YW, SC; Formal Analysis: YL, LW; Funding acquisition: YZ, YW; Methodology: YL, WY, XL, HL, HC; Software: YL, SC; Supervision: YZ; Validation: JZ; Visualization: YL, XL; Writing – original draft: YL; Writing – review & editing: TY, YW, QW, ZW, SC, WL.

*Competing interests.* The authors declare that they have no known competing financial interests or personal relationships that
could have appeared to influence the work reported in this paper.

*Acknowledgments.* Yong Zhao acknowledges support from the National Key Research and Development Program of China (2021YFC3200200), National Natural Science Foundation of China (NSFC) (52025093) and the funds from China Institute of Water Resources and Hydropower Research (IWHR) (WR0145B032021). Yong Wang is supported by the NSFC
(52109044) and IWHR (WR110146B0042024, WR110145B0072024, SKL2024YJTS02).

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
