# Peer review of "Evapotranspiration Stress Intensifies with Enhanced Sensitivity to Soil Moisture Deficits in a Rapidly Greening China"

_EGUsphere, 2024_

## Author Response (AR1)

**Response to Editor**

Dear Prof. Gao,

We gratefully appreciate your prompt handling, encouraging feedback and valuable suggestions for our manuscript.

In response to the constructive comments from both reviewers, we have carefully addressed all points and made comprehensive revisions. The main updates are as follows:

a) **Introduced root-zone soil moisture:** Based on the root depth estimates by Stocker et al. (2023), we calculated the spatiotemporal dynamics of root-zone soil moisture across China over the past two decades and incorporated it into our analytical framework as an influencing factor.

b) **Refined the machine learning analysis:** We improved and re-ran the SHAP model to identify key drivers, and listed performance metrics for both training and validation datasets.

c) **Reorganized the results presentation:** Section 3.3 was rewritten to highlight the distinctions between our MDLM model and conventional methods.

d) **Streamlined the manuscript structure:** We simplified the Methods section by moving detailed calculations to the Appendix, expanded the Discussion section, and added the Conclusion section.

We sincerely hope that these revisions have enhanced the clarity, rigor, and overall quality of the manuscript. All modifications are highlighted in blue in the revised version. We would be grateful if you would consider the updated version for publication, and we truly appreciate your continued support and guidance.

Sincerely,
Yuan Liu (on behalf of all co-authors)

A point-by-point response to your insights is provided as follows.

**EC1:** In this manuscript, you "defined the ratio of actual to potential evapotranspiration, as Evapotranspirative Stress Index (ESI)", and found that China experienced increasing evaporation stress. From my reading, I think your topic is very relevant in ecohydrology, and fits the scope of HESS. You may know that our group emphasized that the root zone is the KEY ecohydrology element, linking atmosphere, vegetation, water, and climate change (Gao et al., 2024; https://doi.org/10.5194/hess-28-4477-2024, 2024.). Moreover, root zone is not static, it is a dynamic zone, changing with climate and human activities (Porporato et al., 2004; Liang et al., 2024). I think the root zone perspective should be more elaborated and discussed in your revision. I'm curious to learn the authors' thoughts on the role of root zone in greening China.

**Reply:** Thank you for your input and guidance. Based on the global rooting depth data from Stocker et al. and ERA5's layered soil moisture, we estimated root-zone soil moisture using a weighted average and analyzed its changes over the past two decades (Appendix

B, Line 616–662). This variable was included in our framework to represent water availability. In Section 3.2, root-zone soil moisture ranked lower in importance for explaining evaporative stress than surface soil moisture (0–7 cm) and precipitation (Figures R2 and R3). We suspect this is due to the use of deseasonalized and detrended data, which may reduce the short-term variability of root-zone moisture. Therefore, we focused on surface soil moisture and precipitation in the subsequent analysis.

The newly added content in Appendix B is as follows (Line 616–662):

[revised manuscript text omitted]

**Figure R3 (Figure S3 in the Supplement). Identification of the key external drivers of ESI.** Same as Fig. 4, but applying the RF regressor. It is consistent with the overall results identified by the XGBoost classifier, except that it fails to distinctly differentiate between surface soil moisture and precipitation.

**EC2:** Line 52-55. Actual evaporation surpassing potential one is an interesting conclusion. I'm thinking whether this conclusion is based on observation or merely modelling. Please elaborate more.

**Reply:** Thank you for pointing this out. Potential evapotranspiration (ETp) is a theoretical value, and its estimation depends on the definition and algorithm used. Different definitions can lead to different results. In this study, we followed the FAO-recommended method, which is based on the Penman equation and assumes a reference crop with uniform surface coverage—specifically, a 10 cm tall alfalfa canopy with associated parameters such as albedo. Under this approach, both model simulations and observations may show actual evapotranspiration exceeding ETp, particularly in humid regions, where vegetation and water availability exceed the assumed reference conditions. Our previous work has documented this phenomenon at the global scale using model outputs (Liu et al., 2022). In the current study, we further compare ESI derived from flux tower observations (actual ET) and ETp calculated via the PM method, and also analyze results from mainstream model products in Appendix A (Line 584-594).

Liu, Y., Jiang, Q., Wang, Q., Jin, Y., Yue, Q., Yu, J., Zheng, Y., Jiang, W., and Yao, X.: The divergence between potential and actual evapotranspiration: An insight from climate, water, and vegetation change, Science of The Total Environment, 807, 150648, https://doi.org/10.1016/j.scitotenv.2021.150648, 2022.

[Figure]

**Figure R4 (Figure A3 in the Appendix A). Spatial distribution of the multi-year average of ESI calculated from ETa and ETp products from different datasets.** (a) was calculated using flux tower observational data, (b) derived from ERA5, (c) originated from GLEAM, (d) obtained from MODIS, and (e) sourced from ERA5 but incorporates the CO2 water-saving effect in ETp calculation.

Relevant text reads (Line 584-594 in Appendix A):

*"Figure A3 compares the ESI derived from three datasets, revealing substantial discrepancies attributable to methodological differences. For ETp estimation, MODIS and ERA5 apply the PM formula calibrated for idealized vegetation types, consistent with our research hypotheses in Figure 1. In contrast, GLEAM integrates corrections for actual vegetation heterogeneity. The ETa < ETp constraint imposed on MODIS and GLEAM limits their ability to capture water surplus conditions (Figure A3cd), whereas ERA5—free of this restriction—produces ESI spatial patterns aligned with eddy covariance (EC) flux measurements (Fig. A3ab). Incorporating $CO_2$ water-saving effects into ETp calculations identifies an expansion of non-water-limited regions, enhancing aridity-humidity characterization ontological congruence (Fig. A3abe). Note that our research is dedicated to analyzing the temporal dynamics of the ESI, particularly in relation to the intensification of drought conditions. The conceptual diagram (Fig. 1) demonstrates that lower ESI values signify increasingly severe imbalances and stress levels. The study does not address how often the threshold of 1 is crossed nor explores the physical significance of this threshold."*

**EC3:** Line 130. decline at a rate of 0.08% d-1. What is the declining rate in mm d-1?
**Reply:** DONE. "d" was originally used for "decade," but as it is not standard, we have replaced it with the full term in all text and figures.

**EC4:** Line 167-170. es, ea, ETp… please use the correct format thoroughly in this paper.
**Reply:** Corrected. We have checked the entire manuscript and revised all subscript and

superscript formatting to ensure consistency and accuracy.

**EC5:** Line 181: CO2, with 2 in lower subscript.
**Reply:** DONE. We apologize for the oversight. We have corrected the figures and texts throughout the main text, Appendix, and Supplement.

**EC6:** Line 190: root zone (0-100 cm) is not a fixed value. The root zone is very diverse in different climate zones (Gao et al., 2014; Wang-Erlandsson et al., 2016; Stocker et al., 2023).10.1002/2014GL061668
DOI: 10.5194/hess-20-1459-2016 20 (4): 1459
https://doi.org/10.1038/s41561-023-01125-2
**Reply:** DONE. Thank you for the input. We have added a section on root-zone soil moisture and cited relevant literature (Line 617-621 in Appendix B).

> *"Previous studies frequently defined the 0-100 cm soil layer as the root zone. However, since the root zone constitutes the most dynamic interface for water-vegetation-atmosphere interactions, employing more realistic root zone soil moisture data as an indicator for evapotranspiration stress is of critical importance (Gao et al., 2014, 2024; Wang-Erlandsson et al., 2016). This research utilizes root depth data corresponding to an 80-year drought recurrence interval, estimated by Stocker et al. (2023) through the mass accumulation curve methodology."*

Gao, H., Hrachowitz, M., Schymanski, S. J., Fenicia, F., Sriwongsitanon, N., and Savenije, H. H. G.: Climate controls how ecosystems size the root zone storage capacity at catchment scale: Root zone storage capacity in catchments, Geophys. Res. Lett., 41, 7916–7923, https://doi.org/10.1002/2014GL061668, 2014.

Gao, H., Hrachowitz, M., Wang-Erlandsson, L., Fenicia, F., Xi, Q., Xia, J., Shao, W., Sun, G., and Savenije, H. H. G.: Root zone in the Earth system, Hydrol. Earth Syst. Sci., 28, 4477–4499, https://doi.org/10.5194/hess-28-4477-2024, 2024.

Stocker, B. D., Tumber-Dávila, S. J., Konings, A. G., Anderson, M. C., Hain, C., and Jackson, R. B.: Global patterns of water storage in the rooting zones of vegetation, Nat. Geosci., 16, 250–256, https://doi.org/10.1038/s41561-023-01125-2, 2023.

Wang-Erlandsson, L., Bastiaanssen, W. G. M., Gao, H., Jägermeyr, J., Senay, G. B., van Dijk, A. I. J. M., Guerschman, J. P., Keys, P. W., Gordon, L. J., and Savenije, H. H. G.: Global root zone storage capacity from satellite-based evaporation, Hydrol. Earth Syst. Sci., 20, 1459–1481, https://doi.org/10.5194/hess-20-1459-2016, 2016.

**EC7:** Line 245: it is fine to only use surnames.
**Reply:** DONE.

**EC8:** Line 379: what is "Hu Line"? I didn't see it in the figures. Please also add a short explanation and reference.

**Reply:** DONE. It has been shown in the study area figure in Appendix A, with relevant explanations provided in the caption (Line 596-604).

[Figure]

**Figure R5 (Figure A1 in the Appendix A). Location of the study area and distribution of multi-year average ESI.** The extent of the study area, employing a tri-chromatic map to differentiate between cropland, forest, and grassland; dots denote the positions of EC flux stations, with colors representing the International Geosphere-Biosphere Programme (IGBP) classifications and shapes indicating observational sources. The inset displays time-series plots of regional average air temperature (red), surface soil moisture (blue), and LAI (green) over the past 20 years, with a black dashed line denoting the linear trend. The red slanted line denotes the position of the "Hu Line (also known as the Hu Huanyong Line)", with its neighboring transitional area defined as the "Hu Zone",a fundamental geographical demarcation in China that characteristically differentiates regional hydrological regimes, vegetation coverage gradients, and demographic distribution patterns (Hu, 1935; Li et al., 2024).

Hu, H. Y.: Essays on China's population distribution, accompanying with statistics and maps (in Chinese with English abstract), Acta Geogr. Sin., 2, 22–74, 1935.

Li, Z., Yang, Q., Ma, Z., Wu, P., Duan, Y., Li, M., and Zheng, Z.: Aridification and Its Impacts on Terrestrial Hydrology and Ecosystems over a Comprehensive Transition Zone in China, Journal of Climate, 37, 1651–1666, https://doi.org/10.1175/JCLI-D-23-0203.1, 2024.

**EC9:** Line 431-432, and Line 477-479: this finding is interesting. But how? And why? I believe ecosystems not only increase their above ground biomass as greening, but also increased their root zone water storage capacity, which increased resilience to droughts.

We found this phenomenon in the US (Liang et al., 2024).
https://doi.org/10.5194/egusphere-2024-550.

**Reply:** DONE. Thank you for your input. We have added a possible explanation of the underlying mechanism in the main text (Line 505-511):

> *"On the other hand, VPD sensitivity and the greening trend exhibit an inverse gradient (Fig. 9d), particularly in arid lands with high $\theta_{VPD}$ values Fig. 8b), indicating that greening in drylands has improved the ability to cope with atmospheric drought. Two mechanisms may explain this phenomenon: (1) Deep-rooted vegetation in certain arid zones accesses deep soil moisture during droughts, releasing it into the atmosphere via transpiration to directly elevate humidity—though this may simultaneously intensify soil aridity (Liang et al., 2024; Sun et al., 2021). (2) Greening lowers surface albedo, suppressing surface temperature increases and thereby mitigating atmospheric drought through indirect thermal regulation (Zhang et al., 2024)."*

Liang, J., Gao, H., Fenicia, F., Xi, Q., Wang, Y., and Savenije, H. H. G.: Widespread increase of root zone storage capacity in the United States, https://doi.org/10.5194/egusphere-2024-550, 11 March 2024.

Sun, G., Gao, H., and Hao, L.: Comments on "Large-scale afforestation significantly increases permanent surface water in China's vegetation restoration regions" by Zeng, Y., Yang, X., Fang, N., & Shi, Z. (2020). Agricultural and Forest Meteorology, 290, 108001, Agricultural and Forest Meteorology, 296, 108213, https://doi.org/10.1016/j.agrformet.2020.108213, 2021.

Zhang, Y., Feng, X., Zhou, C., Sun, C., Leng, X., and Fu, B.: Aridity threshold of ecological restoration mitigated atmospheric drought via land–atmosphere coupling in drylands, Commun Earth Environ, 5, 381, https://doi.org/10.1038/s43247-024-01555-9, 2024.

**EC10:** Line 484-485: we found the same (Sun et al, 2020)
https://www.sciencedirect.com/science/article/pii/S0168192320303154
**Reply:** DONE. Please refer to our response to Comment 9.

**EC11:** Line 492: are do not adhere to. Probably a typo.
**Reply:** DONE. The sentence has been removed in the revised manuscript.

Thank you again for your thorough review and helpful suggestions on root-zone soil moisture, which have greatly improved our manuscript. Please feel free to contact us if anything remains unclear — we truly appreciate your guidance.

**Response to Reviewer #1**

**Reviewer #1 (Remarks to the Author):**

**R1C1:** The article presents a novel method for investigating the driving factors of ecosystem evaporative stress—an area that has already been extensively studied—using multi-source remote sensing data. This method incorporates the self-enhancement mechanism of drought events to achieve more accurate sensitivity estimations. Overall, the paper is well written, with a clearly presented theme, meticulous attention to detail, rigorous conclusions, and notably indicative results.

**Reply:** Thank you very much for your time and expertise in carefully reviewing the manuscript. We appreciate your positive feedback. It is incredibly gratifying to receive such recognition for our work. Below, we provide individual responses to each of your comments.

**R1C1:** The manuscript is overly verbose in some sections. For example, the detailed description of data processing in Section 2 ("Materials and Methods") is unnecessarily elaborate; well-known formulas should be omitted or simplified.

**Reply: DONE.** Following the suggestions from you and Reviewer 2, we have streamlined Section 2 and reorganized the manuscript. The study area description is now in the Introduction (Line 109–116), and all ESI-related calculations has been moved to Appendix A (Line 548–615). A comparison of the structure before and after revision is shown below.

[Figure]

**Figure R1. Comparison of the structure of Section 2 ("Materials and Methods") before and after revision.**

**R1C2:** I argue that, under climate change, global ecosystems will face increasingly severe water scarcity, with soil moisture deficit being a key factor affecting land–atmosphere feedback. While this observation supports the authors' findings, the portrayal of the ESI as representing the "balance" of the water–vegetation–atmosphere system is overemphasized. Although the balance metaphor is vivid, it is not clearly defined; the ESI should be presented modestly as an indicator of ecosystem water deficit.

Reply: DONE. We appreciate your careful review. In our study, we refer to the definition of ETp and its conceptual distinction from ETa. To clarify this, we have added an explanation of the conceptual figure in the Introduction (Figure1, Line 82). We agree with your comment

that the term "equilibrium state" is a figurative expression meant to deepen the interpretation of ESI. As ESI essentially reflects water stress by comparing actual and potential evapotranspiration, we have revised or removed references to the "equilibrium state" in Section 3 ("Results") to maintain scientific rigor. The term is now used only in the discussion to highlight the broader implications of our findings.

Relevant text reads:

Line 265-266:

> *"Consequently, the progression of water stress conditions in these areas necessitates increased scrutiny."*

Line 295-297:

> *"Monthly-scale analysis reveals amplified hydrological regulation (water contribution = 0.56) with concurrent suppression of vegetation and energy controls (Fig. 5b, S4, S5), demonstrating scale-dependent hierarchy in ecosystem stress controls."*

**R1C3:** The manuscript validates the proposed method by comparing results from traditional approaches and different temporal resolutions. It is a solid study. My primary concern is that the MDLM model relies on linear assumptions to quantify sensitivity, despite the results showing time-varying sensitivity in the real world.

**Reply:** We appreciate your concern. Our MDLM model is developed based on the principles of Dynamic Linear Models (DLM), where the estimation of time-varying sensitivity is a key feature. This approach has been previously applied by Zhang et al. (2022) to capture vegetation's dynamic response to precipitation. Our reference to the "real world" is intended to highlight two points: (1) compared to traditional moving-window MLR models, which generalize effects over fixed periods, the Bayesian updating in DLM allows for point-scale, time-specific parameter estimation, better reflecting real-world dynamics; (2) according to differential principles, processes can be approximated as linear over infinitesimally small time intervals at the point scale (as discussed in Liu et al., 2019). The improved simulation performance of our model over MLR also supports this approach.

To clarify the underlying rationale and the distinguishing features of our model, we have added further discussion in Section 4.1 (Line 452–470):

> *4.1 Mechanistic Advancements of the MDLM model*
>
> *……The MDLM model, characterized by its mechanism-driven architecture and operational simplicity, can inherently detrend and deseasonalize data, thus eliminating the need for anomaly pre-processing. It demonstrates significant performance advantages: simulation accuracy is enhanced by 44.26% compared to moving-window MLR (with R² increasing from 0.61 to 0.88, Fig. 6a, S6~S8), consistent with high fitting accuracies (>0.90) reported in other studies (Zhang et al., 2021a). This improvement is primarily attributed to the MDLM's use of Bayesian forward filtering for dynamic parameter estimation and its capability to approximate nonlinear dependencies as quasi-linear relationships at discrete temporal nodes (Fig. 3). Consequently, it provides time-specific analyses, unlike the window-averaged sensitivity generated by MLR (Liu, 2019; Zhang et al., 2022). Pixel-scale model comparisons indicate that the MDLM model consistently exhibits superior performance across the entire study area (Fig. S7, S8). This highlights its high*

*flexibility and broad applicability, enabling adaptation to diverse climatic conditions and terrain features. More importantly, a comparison of the θ sequences for key variables (svm, VPD, and LAI), reveals that the MLR significantly overestimates their temporal trends (Fig. 6~8, S6, S9, S10). These MLR processes—sequence truncation, smoothing, and simplification of nonlinear dynamics—not only lead to signal loss but also cause substantial differences in θ evolution trajectories (Fig. 6g, h; Fig. S6g). These findings highlight the need to re-evaluate the suitability of traditional moving average methods for determining inter-variable coupling strength or temporal sensitivity trends."*

Liu, Y.: Reduced resilience as an early warning signal of forest mortality, Nature Climate Change, 9, https://doi.org/10.1038/s41558-019-0583-9, 2019.

Zhang, Y., Keenan, T. F., and Zhou, S.: Exacerbated drought impacts on global ecosystems due to structural overshoot, Nat Ecol Evol, 5, 1490–1498, https://doi.org/10.1038/s41559-021-01551-8, 2021.

Zhang, Y., Gentine, P., Luo, X., Lian, X., Liu, Y., Zhou, S., Michalak, A. M., Sun, W., Fisher, J. B., Piao, S., and Keenan, T. F.: Increasing sensitivity of dryland vegetation greenness to precipitation due to rising atmospheric CO2, Nat Commun, 13, 4875, https://doi.org/10.1038/s41467-022-32631-3, 2022.

**R1C4:** It remains unclear how these findings might inform improvements in Earth System Models, which typically involve complex, nonlinear interactions. Therefore, the authors should discuss whether alternative nonlinear sensitivity models have been considered or acknowledge the limitations of the linear approach. I suggest that, when highlighting the significance of the study, the phrase "providing a reference for improving Earth system model parameters " be replaced with a more objective description emphasizing the "revelation of time-varying effects."

**Reply:** DONE. Our study quantifies the changes in correlation/coupling/sensitivity between two variables, which are often prescribed as fixed values in ESMs. Therefore, we extended the significance of our work in this context. Following your suggestion, we have revised and streamlined the abstract and conclusion. Now it reads:

Line 36-37:

"*Our findings underscore the spatiotemporal variations in sensitivity, enriching the comprehension of ecosystem reactions to external factors, and offer essential insights for advancing greening endeavors.*"

Line 545-547:

"*Our findings advance mechanistic understanding of complex regional ecosystem dynamics and offer guidance for steering greening strategies toward a more stable equilibrium among water, atmosphere, and vegetation.*"

**Minor comments**

**R1C5:** Line 55: It is recommended to clarify that the $ET_p$ calculation should more accurately be described as reference crop evapotranspiration ($ET_0$), although the two are often used interchangeably.

**Reply:** We agree that these two concepts are often used interchangeably and have clarified them in the caption of the conceptual figure. Following your suggestion, we have also added clarification in the Introduction to avoid potential ambiguity.

Line 50:

> "*Potential evapotranspiration (ETp, also known as ET0, reference crop evapotranspiration in agriculture) represents the evaporation potential under ideal circumstances—ample water supply, uniform plant growth, and consistent crop coverage (specifically, alfalfa)—and generally considered to be influenced solely by climatic factors (Allen et al., 1998; Li et al., 2022; Thornthwaite, 1948).*"

Allen, R. G., Pereira, L. S., Raes, D., and Smith, M.: Crop evapotranspiration : guidelines for computing crop water requirements, 1998.

Li, Y., Qin, Y., and Rong, P.: Evolution of potential evapotranspiration and its sensitivity to climate change based on the Thornthwaite, Hargreaves, and Penman–Monteith equation in environmental sensitive areas of China, Atmospheric Research, 273, 106178, https://doi.org/10.1016/j.atmosres.2022.106178, 2022.

Thornthwaite, C. W.: An Approach toward a Rational Classification of Climate, Geographical Review, 38, 55, https://doi.org/10.2307/210739, 1948.

**R1C6:** Lines 72, 76, 78, 100: The punctuation is not standardized; for example, a space should be inserted after parentheses in citations. A thorough check throughout the manuscript is advised.

**Reply:** Done. Thank you for pointing this out. We have carefully checked the manuscript.

**R1C7:** Line 89:  Replace "variations" with "differences" in the phrase "Variations in regional…" to more accurately convey the intended meaning.

**Reply: DONE.** Now it reads:

Line 90:

> "*Differences in regional evaporation stress patterns arise from different response to relevant factors.*"

**R1C8:** Line 116:  The term "high-frequency analysis" is unclear; please either provide clarification or remove the term.

**Reply:** DONE. This part has been removed, and the sentence has been revised to:

Line 125:

> "*Our findings are expected to narrow the knowledge gap in temporal variability of evapotranspiration stress response,*"

**R1C9:**Lines 172–185:(1)Is the formula accounting for the $CO_2$ water-saving effect widely accepted? Are there studies that validate its accuracy?(2)Are subsequent analyses based entirely on the $CO_2$-adjusted ESI? This comparison is only made in Section 3.1; clarification in other sections is needed.

**Reply:** First, regarding your Comment 1, this formula is widely accepted and commonly used. We have reviewed recent studies that adopted this formula, and have added clarifying statements and corresponding references to demonstrate its recognition.

We have added clarifying statements and relevant references to support the acceptance of this formula (Line 134-136):

> *"To avoid potential overestimation of ETp, we implement the Yang et al., (2019)-modified PM formula, which incorporates the $CO_2$ water-saving effect and is widely adopted in evapotranspiration studies (Lian et al., 2021; Zhang et al., 2023)."*

We have also added a description in Appendix A (Line 575-577):

> *"This formula has been widely recognized and applied in recent studies, providing an important theoretical basis for accurately estimating evapotranspiration (Lian et al., 2021; Zhang et al., 2023)"*

Regarding Comment 2, following your suggestion, we have added the following content at the end of Section 2 ('Materials and Methods') to clarify the research process.
Line 239-242:

> *"This study comprehensively investigates the dynamic trajectory of evaporative stress by comparing the Evaporative Stress Index (ESI) calculated using the conventional PM formula from 1950 to 2020 with the ESI derived from the PM formula incorporating $CO_2$ water-saving effects from 2001 to 2020, enabling a detailed examination of its variation characteristics. Subsequent analyses are conducted using the latter due to limitations in the influencing factors' data series."*

Lian, X., Piao, S., Chen, A., Huntingford, C., Fu, B., Li, L. Z. X., Huang, J., Sheffield, J., Berg, A. M., Keenan, T. F., McVicar, T. R., Wada, Y., Wang, X., Wang, T., Yang, Y., and Roderick, M. L.: Multifaceted characteristics of dryland aridity changes in a warming world, Nat Rev Earth Environ, 2, 232–250, https://doi.org/10.1038/s43017-021-00144-0, 2021.

Yang, Y., Roderick, M. L., Zhang, S., McVicar, T. R., and Donohue, R. J.: Hydrologic implications of vegetation response to elevated CO2 in climate projections, Nature Clim Change, 9, 44–48, https://doi.org/10.1038/s41558-018-0361-0, 2019.

Zhang, Y., Zheng, H., Zhang, X., Leung, L. R., Liu, C., Zheng, C., Guo, Y., Chiew, F. H. S., Post, D., Kong, D., Beck, H. E., Li, C., and Blöschl, G.: Future global streamflow declines are probably more severe than previously estimated, Nat Water, https://doi.org/10.1038/s44221-023-00030-7, 2023.

**R1C10:** Line 245: The citation "work of Yanlan Liu, (2019)" requires correction in its formatting.

**Reply: DONE.** Thanks.

**R1C11:** Some border lines in Figure 3 impair readability; please check and remove them as needed.

**Reply:** DONE. We apologize for the oversight. Following your Comment #17 and the suggestion from another anonymous reviewer, we have redrawn all figures in the main text, Appendices, and Supplement. The main changes include:

(1) For maps, we removed internal grids, added scale bars, and integrated color bars to improve clarity and compliance with standard practices.

(2) Font sizes of labels and numbers have been increased for better readability.

(3) Resolution has been enhanced to improve image quality.

We hope these revisions make the figures easier to follow and more effective in conveying key information.

**R1C12:** Line 317: The phrase "consistent with Liebig's law of the minimum" should be accompanied by an appropriate reference.

**Reply: DONE.** Now it reads:

Line 286-288:

> *"especially in water-limited regions, consistent with Liebig's law of the minimum (Danger et al., 2008; Tang and Riley, 2021)."*

Danger, M., Daufresne, T., Lucas, F., Pissard, S., and Lacroix, G.: Does Liebig's law of the minimum scale up from species to communities?, Oikos, 117, 1741–1751, https://doi.org/10.1111/j.1600-0706.2008.16793.x, 2008.

Tang, J. and Riley, W. J.: Finding Liebig's law of the minimum, Ecological Applications, 31, e02458, https://doi.org/10.1002/eap.2458, 2021.

**R1C13:** Line 345: Why soil moisture in the 100–289 cm depth range shows higher relative importance compared to other depths in Figure 6?

**Reply:** We thank the reviewer for this insightful comment. Based on our literature review, we speculate that this phenomenon may be related to the vertical partitioning of ecosystem water use: surface moisture dominates short-term stress responses, while deep-layer moisture regulates long-term stress, particularly in dry seasons or arid regions (Fang et al., 2016; Li et al., 2021). The middle layer, influenced by both upper and lower layers and characterized by short water residence time, may contribute less due to its high variability (Huang et al., 2016). In addition, our use of detrended and deseasonalized variables may have amplified the apparent contribution of deep-layer moisture to long-term stress. We acknowledge that this is only a hypothesis, as we could not identify direct supporting

studies. Further research on this topic is warranted. We appreciate the reviewer's guidance.

Fang, X., Zhao, W., Wang, L., Feng, Q., Ding, J., Liu, Y., and Zhang, X.: Variations of deep soil moisture under different vegetation types andinfluencing factors in a watershed of the Loess Plateau, China, Hydrol. Earth Syst. Sci., 20, 3309–3323, https://doi.org/10.5194/hess-20-3309-2016, 2016.

Huang, X., Shi, Z. H., Zhu, H. D., Zhang, H. Y., Ai, L., and Yin, W.: Soil moisture dynamics within soil profiles and associated environmental controls, CATENA, 136, 189–196, https://doi.org/10.1016/j.catena.2015.01.014, 2016.

Li, B.-B., Li, P.-P., Zhang, W.-T., Ji, J.-Y., Liu, G.-B., and Xu, M.-X.: Deep soil moisture limits the sustainable vegetation restoration in arid and semi-arid Loess Plateau, Geoderma, 399, 115122, https://doi.org/10.1016/j.geoderma.2021.115122, 2021.

**R1C14:** Lines 360–365: Attributing the soil moisture sensitivity peak in 2016 solely to extreme climatic conditions may be one factor, but I believe it is unlikely to be the primary reason. Many drought indicators exhibit a unimodal pattern peaking around 2010, suggesting that complex interactions among various ecosystem variables may be involved.

**Reply:** We fully agree with the reviewer—our previous explanation of this phenomenon was indeed incorrect. On one hand, both methods we used consistently revealed a peak in soil moisture sensitivity to evaporative stress around 2015. As you noted, similar trends have been reported in multiple studies, supporting the robustness of our results. In addition, we found that Tang et al. observed a similar increase-then-decrease pattern in vegetation sensitivity to soil drought, and attributed this shift mainly to climate change based on controlled simulations with 12 TRENDY models. This study supports our findings and offers a possible explanation. We have revised the main text accordingly.

Relevant text reads (Line 349-355):

> *"Soil moisture stands out as the paramount and most sensitive factor, with its sensitivity demonstrating a significant upward trend in both models, albeit with quantitative differences (Fig. 6b). According to the MDLM results, $\theta_{svm}$ increased by 9.49% over the past two decades (0.03±0.01 yr$^{-1}$, p < 0.001), peaking in 2016, which represents a 13% rise compared to the stable level of the previous decade. In contrast, the MLR model, while producing a slightly lower mean $\theta_{svm}$ (2.36 vs. 2.98), exhibited a more pronounced growth trend (0.05±0.01 yr$^{-1}$, p < 0.001), with the peak occurring earlier in 2014. This shift in peak timing is an inherent consequence of the moving average process. As observed in similar studies, Tang et al. (2025) reported analogous evolutionary patterns and attributed this phenomenon primarily to climatic drivers."*

Tang, J., Niu, B., Fu, G., Peng, J., Hu, Z., and Zhang, X.: Shifted trend in drought sensitivity of vegetation productivity from 1982 to 2020, Agricultural and Forest Meteorology, 362,

110388, https://doi.org/10.1016/j.agrformet.2025.110388, 2025.

**R1C15:** The reported standard deviation in "…with an estimated 4.43% (±6.88%) increase…" is notably large, even exceeding the mean. Please explain this occurrence.

**Reply:** We apologize for the confusion. The original standard deviation referred to the values across spatial grids, which was reasonable given the substantial variation in sensitivity across different regions. To better highlight the core focus of our study—'temporal variation in sensitivity'—we have revised the presentation. Figure 6 now shows the annual trend of average sensitivity across the entire study area. Accordingly, we have updated the statistics in the main text to reflect the multi-year average for the whole region. Figure 6 has been revised (also see below Figure R2), along with the corresponding text (Line 356-359).

Relevant text reads (Line 356-359):

> "On average, a 1 kPa rise in VPD correlates with an estimated approximate 16.84% rise in evaporative stress based on the MDLM, which is slightly lower than the 19.74% increase estimated by the MLR. Moreover, the MLR predicts a higher annual growth rate (0.37 ± 0.07% yr$^{-1}$ vs 0.24 ± 0.05% yr$^{-1}$, p < 0.001) (Fig. 6c). "

[Figure]

**Figure R2. Sensitivity of the ESI to external drivers and its dynamic trajectory.** (a) Distributions of R², RMSE, and MAE for MLR and MLDM models across all grid cells in training and validation sets, with triangles marking median positions. (b–i) $\theta$ time series derived from MLDM (red) and MLR (blue). Dashed

lines indicate linear or quadratic fit trends, with shaded regions representing 95% confidence intervals of the fits. Text annotations display Sen's slope estimates with statistical significance, along with R² values and fit significance; quadratic fits were applied when linear trends were insignificant. Boxplots illustrate $\theta$ distributions: boxes denote interquartile ranges (25th–75th percentiles), horizontal lines represent medians, dots indicate means, whiskers span 5th–95th percentiles, and text annotations specify t-test significance of differences between model results.

**R1C16:** Line 410: In Figure 8g, why does southern China (e.g., the lower Yangtze region) appear magenta, and why does evaporative stress increase despite an increase in LAI? A detailed explanation is required.

**Reply:** This is a very insightful comment. We have closely re-examined the result (now shown as Figure 7c in the revised version) and have added a detailed description and discussion of this phenomenon in the text.

Relevant text reads (Line 385-391):

> "Approximately 29.6% of the national territory (2.81 ×10⁶ km²), counterintuitively, experienced an increase in LAI coupled with intensified ET stress ($\theta_{LAI}$ < 0), predominantly in the Loess Plateau, North China Plain, and lower Yangtze River region ( Fig. 7c). In the Loess Plateau and North China Plain, extensive research has linked rapid greening and the expansion of intensive agriculture to heightened water deficits (Feng et al., 2016; Lu et al., 2018; Wang et al., 2024). In the lower Yangtze River region, observational evidence suggests that elevated LAI reduces evaporation (Zhang et al., 2021c). This phenomenon may arise from the shading effect of increased LAI in energy-limited areas, which diminishes radiation reaching the surface, lowers soil temperature, and thereby reduces soil evaporation (Forzieri et al., 2020). "

Feng, X., Fu, B., Piao, S., Wang, S., Ciais, P., Zeng, Z., Lü, Y., Zeng, Y., Li, Y., Jiang, X., and Wu, B.: Revegetation in China's Loess Plateau is approaching sustainable water resource limits, Nature Clim Change, 6, 1019–1022, https://doi.org/10.1038/nclimate3092, 2016.

Forzieri, G., Miralles, D. G., Ciais, P., Alkama, R., Ryu, Y., Duveiller, G., Zhang, K., Robertson, E., Kautz, M., Martens, B., Jiang, C., Arneth, A., Georgievski, G., Li, W., Ceccherini, G., Anthoni, P., Lawrence, P., Wiltshire, A., Pongratz, J., Piao, S., Sitch, S., Goll, D. S., Arora, V. K., Lienert, S., Lombardozzi, D., Kato, E., Nabel, J. E. M. S., Tian, H., Friedlingstein, P., and Cescatti, A.: Increased control of vegetation on global terrestrial energy fluxes, Nat. Clim. Chang., 10, 356–362, https://doi.org/10.1038/s41558-020-0717-0, 2020.

Lu, C., Zhao, T., Shi, X., and Cao, S.: Ecological restoration by afforestation may increase groundwater depth and create potentially large ecological and water opportunity costs in arid and semiarid China, Journal of Cleaner Production, 176, 1213–1222, https://doi.org/10.1016/j.jclepro.2016.03.046, 2018.

Wang, Q., Liu, H., Liang, B., Shi, L., Wu, L., and Cao, J.: Will large-scale forestation lead to a soil water deficit crisis in China's drylands?, Science Bulletin, 69, 1506–1514, https://doi.org/10.1016/j.scib.2024.03.005, 2024.

Zhang, Y., Kong, D., Zhang, X., Tian, J., and Li, C.: Impacts of vegetation changes on global evapotranspiration in the period 2003-2017, Acta Geographica Sinica, 76, https://doi.org/10.11821/dlxb202103007, 2021.

**R1C17:** General Comment on Figures: The text in the figures is not sufficiently clear. It is strongly recommended to increase the resolution of all figures to improve readability.
**Reply:** DONE. All figures have been redrawn. Please see our reply to Comment 11 for details.

Thank you again for your thorough review and positive feedback. Please do not hesitate to contact us if any clarification or further revision is needed.

**Response to Reviewer #2**

**Reviewer #2 (Remarks to the Author):**

**General Comments:** The authors examine the monotonic trend of evapotranspiration stress (measured by ESI). They investigate the sensitivity of ESI and its changes to the hydrological, atmospheric, and vegetation variables across China. They employ explainable machine learning methods to discern key variables that determine the ESI. They then establish another novel regression model with time varying parameters to investigate the sensitivity of the controlling factors. The authors find a predominant role of soil moisture to ESI changes. The feedback loop of the water-vegetation-atmosphere the authors discussed is interesting. The authors did extensive data-driven analyses, and their findings could inform the land use policies. I am generally supportive of publishing the paper, once below comments about the wording, methods, mapping and structure are clarified or addressed.

**Reply:** Thank you for your careful evaluation of our work and constructive comments on our manuscript. We appreciate the insights you have shared, as they have been instrumental in further improving the quality of our study.

Based on your suggestions, we have made extensive revisions to improve the clarity of the manuscript and tighten the presentation of our findings. Specifically: (1) Section 3.2 on machine learning has been reworked to improve validation accuracy; (2) Section 3.3 on temporal sensitivity has been rewritten to better highlight the comparison between the MDLM and MLR models; (3) the Conclusion has been added and the Discussion expanded to sharpen the focus. In addition, we have streamlined the Methods section to enhance readability. We hope these revisions make the manuscript easier to follow and offer new insights into the dynamics of the water-energy-atmosphere nexus in ecosystems.

Below, we respond to each of your comments in detail.

**R2C1:** Title Clarity: The title may be misleading. While the authors emphasize the increasing sensitivity of ESI to soil moisture, it is important to note that both the sensitivity ($\theta$ svm) and the actual soil moisture ($\delta$ svm) are changing and contributing to ESI changes (Figures 5-6). It is unclear whether the monotonic trend ($\delta$ change) or the variability changes ($\theta$ change) are more dominant in driving ESI changes. Therefore, the title "Increasing sensitivity … predominantly intensifies …" might be misleading.

**Reply: DONE.** This is an excellent suggestion that helped clarify our logic. Thank you for pointing it out. Following your advice, we revised both the title and relevant expressions in the main text to avoid misleading.

The original title:

> *"Increasing Sensitivity to Soil Moisture Deficits Predominantly Intensifies Evapotranspiration Stress in a Greening China"*

has been revised to:

> *"Evapotranspiration Stress Intensifies with Enhanced Sensitivity to Soil Moisture Deficits in a Rapidly Greening China"*

Relevant text reads (Line 529-532):

*"Finally, what are the key implications given the foreglimpse of intensifying evapotranspiration stress coinciding with the increasing sensitivity to soil water deficits in a greening China? Soil water variability would lead to greater changes in the ecosystems undergoing rapid greening. That is, stronger drought effects can be expected when soil water is anomalously low."*

**R2C2:** Temporal Resolution Consistency:  The connection between Section 3.1 and other sections appears weak. Figures 4a and 4c depict the ESI trend, but it is unclear whether this represents the annual mean or another temporal scale. The regression models (RF, XGBoost, MDLM) are built at monthly and sub-monthly scales. The authors need to reconcile these inconsistencies by clarifying the temporal resolutions of the ESI and sensitivity trends.

**Reply: DONE.** First, our study uses the ESI to characterize ecosystem water stress and aims to investigate its temporal evolution, as well as the changing sensitivity of ESI to its driving factors. Section 3.1 addresses the first research question, forming the foundation of the study and aligning with the manuscript title. In Section 3.2, we applied the SHAP model to identify the main drivers of ESI, while Sections 3.3 and 3.4 quantify ESI sensitivity to these drivers using the MDLM and MLR models, respectively. This forms the narrative structure of our study.

We appreciate your comment regarding the temporal scale used throughout the manuscript. The original version lacked consistency, and we have now revised Section 3.3 with updated statistical analyses to ensure all trends are evaluated at the interannual scale. This also accounts for revisions to some of the statistical results in the manuscript. To clarify this point, we have added the following statement at the end of Section 2 (Materials and Methods, Line 239-245):

*"This study comprehensively investigates the dynamic trajectory of evaporative stress by comparing the Evaporative Stress Index (ESI) calculated using the conventional PM formula from 1950 to 2020 with the ESI derived from the PM formula incorporating $CO_2$ water-saving effects from 2001 to 2020, enabling a detailed examination of its variation characteristics. Subsequent analyses are conducted using the latter due to limitations in the influencing factors' data series. The SHAP model is applied at 8-day, 16-day, and monthly scales to obtain temporally robust importance rankings. Furthermore, to calculate sensitivity, the MDLM model operates at an 8-day temporal resolution, as a larger sample size enhances parameter accuracy. The temporal evolution patterns of sensitivity coefficients are explored at an interannual scale."*

**R2C3:** The Model Comparison:  The authors claim that the MDLM model is novel and superior to the MLR method based on inconsistent results (Lines 393-394) and higher $R^2$ values (Line 490). However, an in-depth comparison of the sensitivity and its changing patterns derived from both methods is lacking. Additionally, the authors should specify the names of the "two distinct methodologies" mentioned in Line 394.

**Reply: DONE.** Thank you for your constructive suggestion. We have revised the

presentation of all results in Section 3.3. While our initial intention was to highlight the improved performance of the MDLM model, we now provide a comparative analysis of both models in terms of accuracy and result differences. This revision better demonstrates the innovation and advantages of the developed model. Please see Figs. 6–7 and Lines X–X (covering most of Section 3.3) in the revised manuscript, as well as Figs. S6–S10 in the Supplement.

[Figure]

**Figure R1.(Figure 2 in main text, Line 404). Sensitivity of the ESI to external drivers and its dynamic trajectory.** (a) Distributions of R², RMSE, and MAE for MLR and MLDM models across all grid cells in training and validation sets, with triangles marking median positions. (b–i) $\theta$ time series derived from MLDM (red) and MLR (blue). Dashed lines indicate linear or quadratic fit trends, with shaded regions representing 95% confidence intervals of the fits. Text annotations display Sen's slope estimates with statistical significance, along with R² values and fit significance; quadratic fits were applied when linear trends were insignificant. Boxplots illustrate $\theta$ distributions: boxes denote interquartile ranges (25th–75th percentiles), horizontal lines represent medians, dots indicate means, whiskers span 5th–95th percentiles, and text annotations specify t-test significance of differences between model results.

[Figure]

**Figure R2.(Figure 7 in main text, Line 412). Spatiotemporal heterogeneity in sensitivity of ESI to svm0-7, VPD, and LAI.** (a–c) Spatial distributions of temporal trends (estimated by Sen's slope) for $\theta_{svm}$ (sensitivity of ESI to svm0-7), $\theta_{VPD}$, and $\theta_{LAI}$, respectively. Orange indicates increasing trends, blue denotes decreasing trends, with hatched patterns marking significance at $p < 0.05$ (Mann-Kendall test). Insets display spatial distributions of multi-year mean $\theta$.(d–f) Discrepancies in Sen's slope estimates and multi-year mean values between MLR- and MDLM-derived $\theta$ (a–c). Pie charts quantify the proportion of land area exhibiting corresponding trend patterns. Gray indicates non-vegetated areas.

[Figure]

**Figure R3. (Figure S6 in Supplement). Sensitivity of the ESI to pivotal external drivers and its dynamic trajectory**. Same as Fig. 6, but here the models establish the sensitivity profile for the following external factors: Precipitation -Prec, Rn, T, VPD, U, $CO_2$, and LAI (Parameter Set: R1). In Figure a, the MDLM model demonstrated significantly higher accuracy in both the training and validation sets compared to MLR. The sequences of Prec, VPD, LAI, Rn, and T derived from the two models exhibited substantial differences, highlighting the MDLM model's notable improvement in predicting key variables. Except for Rn, the sensitivity of additional factors displayed a consistent pattern in both magnitude and directional trend. The observed trend of key factors aligns with the R0 scenario that corroborates the robustness of research findings. These findings indicate heightened sensitivity to both water supply and VPD, coupled with a diminished sensitivity of LAI.

[Figure]

**Figure R4. (Figure S7 in Supplement). Simulation accuracy and validation of the MDLM model.** For each pixel, the dataset was partitioned into a training set and a validation set at a ratio of 8:2, facilitating the evaluation of the model's ability to accurately fit the actual ESI sequence, along with its local, seasonal, and trend components. This process is exemplified through the analysis of a single pixel in (g), with the method detailed in Text S1. (a-c) The distribution of the $R^2$, RMSE, and MAE for the training set, with R0 parameter set. These figures include insets that show the distributions across the region, with the mean and median values delineated by red and green dashed lines, respectively. (d-f) The model's performance for the validation set. The analyses reveal that the MDLM model achieves a consistently high fitting accuracy, evidenced by an average $R^2$ of 0.88, an RMSE of 0.04, and an MAE of 0.03. It is noted that results exhibit suboptimal performance in certain areas of the southeastern region, however, remain relatively high credibility in other areas.

[Figure]

**Figure R5. (Figure S8 in Supplement). Simulation accuracy and validation of the MLR model.** (a-c) The distribution of the $R^2$, RMSE, and MAE for model with Parameter Set R0. The model achieved an average $R^2$ of 0.60, RMSE of 0.13, and MAE of 0.09 across the entire region. (d-f) The model's performance using the Parameter Set R1, with an average $R^2$ of 0.59, RMSE of 0.13, and MAE of 0.09 across the entire region.

[Figure]

**Figure R6. (Figure S9 in Supplement). Spatiotemporal heterogeneity in svm0-7, VPD, and LAI sensitivity.** Same as Fig. 7a-c, but the sensitivity is derived from the MLR model with the external driving factors input (R0): svm0-7, Rn, T, VPD, U, $CO_2$, and LAI. Given that the MLR model generates an averaged sensitivity series within a 5-year moving window, and the MLDM produces time-specific sequences, the figure is not numerically comparable to Figure 8. Instead, it serves to contrast the spatial distribution patterns and relative magnitudes across different categories. The patterns for $\theta_{svm}$ $\theta_{VPD}$ and $\theta_{LAI}$ depicted in figures (a-f) are consistent with Fig. 7, with similar area proportions, affirming the results' robustness. Here, the same color scheme as Fig. 7 (a-f) is used, which also reveals the MLR model's significant overestimation of sensitivity trend.

[Figure]

**Figure R7. (Figure S10 in Supplement). Spatiotemporal heterogeneity in sensitivity of ESI to Prec.** Same as Fig. 9a,Fig. S9a and Fig. 9d, but the sensitivity is derived from the MDLM(a) and MLR(b) model with the external driving factors input (R1): Prec, Rn, T, VPD, U, $CO_2$, and LAI. The main figure displays the spatial distribution of the Sen's slope for $\boldsymbol{\theta_{Prec}}$, whereas the inset presents the spatial distribution of the multi-year average of $\boldsymbol{\theta_{Prec}}$. Figure (c) delineates the differences between (a) and (b). This figure further corroborates the analogous patterns in the simulation outcomes of the two models, along with the MLR model's overestimation of the intensity of change trends, where both positive and negative trends are more pronounced.

Based on the revised figures, we have rewritten Section 3.3. As the content is substantial, we kindly refer you to the main text (Line 333-422) for details.

In addition, regarding your suggestion to specify the "two distinct methodologies" mentioned in Line 394, we have revised the sentence for clarity. It now reads as follows (Line 396-400):

> *"Figure 8 further illustrates that the values and trends of $\theta_{svm}$, $\theta_{VPD}$, and $\theta_{LAI}$, as estimated by MLDM and MLR, display statistically significant differences (p < 0.001) across nearly all climate zones and vegetation types, underscoring substantial methodological discrepancies. Within the same land use categories, the MLR model consistently demonstrates more dispersed sensitivity trend distributions and greater intra-group variability (Fig.8)."*

More importantly, based on these revisions, we conducted a more careful comparison of the spatial patterns of sensitivity. Compared with the original manuscript, we have identified

a new finding: the sensitivities to precipitation and soil moisture exhibit complementary spatial patterns (Figure R2a, R7a; corresponding to Figure 7a and Figure S10a in the manuscript). This observation has been added to both the Results and Discussion sections.

Line 364–367 in Results section:

  *"Spatially, $\theta_{svm}$ is predominantly higher in the southeast and lower in the northwest, whereas $\theta_{Prec}$ exhibits a complementary pattern (Fig. 7a, S10a). This indicates that in humid regions, the ESI is more significantly affected by soil moisture regulation, while in arid regions, short-term precipitation can substantially alleviate evaporative stress. The specific mechanisms are discussed in detail in section 4.2."*

Lines 485–492 in Discussion section:

  *"Intriguingly, our research identifies a complementary spatial pattern between $\theta_{svm}$ (southern high-northern low) and $\theta_{Prec}$ across China (Fig. 7a, S10a). In northern arid/semi-arid regions characterized by low initial soil moisture, precipitation acts as an "instantaneous input" that rapidly replenishes soil moisture and substantially alleviates evaporative stress, exhibiting higher sensitivity. Conversely, in southern humid regions where soils maintain near-saturation moisture levels, surpassing the pre-existing hydro-equilibrium threshold triggers disproportionately severe ecosystem perturbations. Consequently, ESI in these regions demonstrates heightened dependence and sensitivity to soil moisture dynamics."*

**R2C4:** Materials and Methods Structure: The Materials and Methods section is overwhelming. The subsection 2.1.1 "Study Area" could be moved to the Introduction section as background information. Given the extensive list of datasets and variables used, with various spatiotemporal scales, sources, and pre-processing procedures, I suggest the authors compile this information into tables for better readability.

**Reply: DONE.** Thanks for your suggestion. In response, and in alignment with Reviewer1's comments, we have streamlined Section 2 ("Materials and Methods") through the following adjustments:

(1) The study area description has been moved to the Introduction (Line 109–116);

(2) The full content related to ESI calculation—including textual descriptions, figures, and results—has been relocated to Appendix A (Line 548–615);

(3) Additional supporting analyses and procedures are now included in Appendices B and C (Line 616-686);

(4) Information on variables, datasets, and preprocessing steps has been compiled into Table S1 (in Supplementary Materials).

A comparison of the section structure before and after revision is provided below:

[Figure]

**Figure R8. Comparison of the structural adjustments made to Section 2 ("Materials and Methods").**

The study area description in the Introduction is now provided as follows (Line 109–116):

*"Over the past two decades, China's mean air temperature has increased at a rate of 0.24°C per decade (decade $^{-1}$), exceeding the global average of 0.20°C decade $^{-1}$.Concurrent with climate warming and large-scale ecological restoration initiatives—including the Three-North Shelter Forest Program, Grain for Green Program, and Plain Greening Project (Fu et al., 2024a), the regional leaf area index (LAI) has risen by 7.66% (4.21% decade $^{-1}$, p < 0.001), the highest rate globally (Chen et al., 2019). However, surface soil moisture has simultaneously declined at 0.08% decade $^{-1}$, signaling hydrological trade-offs. These rapid environmental transformations, driven by both climate variability and anthropogenic activities, have established China as a critical natural laboratory for studying coupled hydro-ecological-climatic interactions (Bai et al., 2020; Li et al., 2018; Zheng et al., 2022)"*

Bai, P., Liu, X., Zhang, Y., and Liu, C.: Assessing the Impacts of Vegetation Greenness Change on Evapotranspiration and Water Yield in China, Water Resources Research, 56, e2019WR027019, https://doi.org/10.1029/2019WR027019, 2020.

Chen, C., Park, T., Wang, X., Piao, S., Xu, B., Chaturvedi, R. K., Fuchs, R., Brovkin, V., Ciais, P., Fensholt, R., Tømmervik, H., Bala, G., Zhu, Z., Nemani, R. R., and Myneni, R. B.: China and India lead in greening of the world through land-use management, Nat Sustain, 2, 122–129, https://doi.org/10.1038/s41893-019-0220-7, 2019.

Fu, F., Wang, S., Wu, X., Wei, F., Chen, P., and Grünzweig, J. M.: Locating Hydrologically Unsustainable Areas for Supporting Ecological Restoration in China's Drylands, Earth's Future, 12, e2023EF004216, https://doi.org/10.1029/2023EF004216, 2024.

Li, Y., Piao, S., Li, L. Z. X., Chen, A., Wang, X., Ciais, P., Huang, L., Lian, X., Peng, S., Zeng, Z., Wang, K., and Zhou, L.: Divergent hydrological response to large-scale afforestation and vegetation greening in China, Sci. Adv., 4, eaar4182, https://doi.org/10.1126/sciadv.aar4182, 2018.

Zheng, H., Miao, C., Li, X., Kong, D., Gou, J., Wu, J., and Zhang, S.: Effects of Vegetation Changes and Multiple Environmental Factors on Evapotranspiration Across China Over the Past 34 Years, Earth's Future, 10, e2021EF002564, https://doi.org/10.1029/2021EF002564, 2022.

The compiled table in the appendix is as follows:

Table S1. Summary of the dataset applied in this study.

| Role | | Variable | Abbreviation | Unit | Source/Obtained Process | Spatial Resolution and Range | Temporal Resolution and Span |
|---|---|---|---|---|---|---|---|
| Compute ESI | | Actual Evapotranspiration | ETa | mm | ERA5 | 0.1°, Global | Daily, 1950-2020 |
| | | Potential Evapotranspiration | ETp | | | | |
| | | ETp considering the $CO_2$ water-saving effect | ETpPMCO$_2$ | mm | Calculated by Eq.(A4). | | |
| As Influencing Factors | Water supply | Total Precipitation | Prec | mm | ERA5 | 0.1°, Global | Daily, 2001-2020 |
| | | Multi-layer Soil Moisture | svm0-7; svm7-28, svm28-100, svm100-289 | $m^3 \cdot m^{-3}$ | | | |
| | | Root-zone Soil Moisture | Rsvm | $m^3 \cdot m^{-3}$ | Calculated using Eq. Bx. | | |
| | Energy Conditions | 2m Air Temperature | T | °C | ERA5 | 0.1°, Global | Daily,2001-2020 |
| | | Net Radiation | Rn | $MJ \cdot m^{-2}$ | Calculated by summing net solar radiation and net thermal radiation from ERA5. | | |
| | Atmospheric Conditions | Atmospheric Pressure | P | kPa | ERA5 | 0.1°, Global | Daily, 2001-2020 |
| | | $CO_2$ Mole Fraction | $CO_2$ | ppm | CT2022 | 3×2°, Global | 3-hourly, 2001-2020 |
| | | 2m Wind Speed | U | $m \cdot s^{-2}$ | 10m wind speed is calculated using the Pythagorean theorem with ERA5 10m u- and v-components of wind, then converted to 2m wind speed with the formula: $U = U_{10} * \frac{4.87}{\ln(67.8*10-5.42)}$. | | |
| | | Vapor Pressure Deficit | VPD | kPa | Calculated based on the Clausius-Clapeyron relation using 2m temperature and dew point temperature from ERA5 (Held and Soden, 2006; Zhong et al., 2023). | | |
| | Vegetation Status | Leaf Area Index | LAI | $m^2 \cdot m^{-2}$ | GLASS V60 | 500m, Global | 8-day, 2001-2020 |
| | | Fractional Vegetation Cover | FVC | Dimensionless | GLASS V40 | | |
| | | Normalized Difference Vegetation Index | NDVI | Dimensionless | MOD13C1 | 0.05°, Global | 16-day, 2001-2020 |
| | | Enhanced Vegetation Index | EVI | | | | |
| | | kernel NDVI | kNDVI | Dimensionless | Calculated based on NDVI data using Eq.(2). | | |
| Auxiliary Data | | Aridity Index | AI | Dimensionless | Calculated by the ratio of annual Prec to ETp from the ERA5 dataset. | | |
| | | China Land Cover Dataset | - | - | Yang and Huang, (2021) | 30m, China | Annual, 2001-2020 |
| | | Root Depth | - | mm | Stocker et al. (2023) | 0.05°, Global | - |

**R2C5:** Map Issues: The gridline spacing in Figures 2b and 2c should be consistent. I recommend using a narrower legend and specifying kilometers as the unit for clarity.

**Reply: DONE.** Thank you for your helpful suggestions on figure improvement. In response, we have made the following adjustments to all figures in the main text, Appendix, and Supplement:

(1) For maps, we removed the internal latitude-longitude grid, added scale bars (in kilometers), and moved color bars inside the map panels to enhance clarity and adhere to standard cartographic practices.

(2) We increased the font size of labels and numbers to improve readability.

(3) We enhanced the image resolution to improve overall figure quality.

We hope these revisions make the figures easier to interpret and more effective in conveying the key findings.

The revised Figures 2b and 2c are now presented as Figure A3 (Line 613) in Appendix A.

[Figure]

**Figure R9** (**Figure A1 in Appendix A, Line 613**). **Spatial distribution of the multi-year average of ESI calculated from ETa and ETp products from different datasets.** (a) was calculated using flux tower observational data, (b) derived from ERA5, (c) originated from GLEAM, (d) obtained from MODIS, and (e) sourced from ERA5 but incorporates the $CO_2$ water-saving effect in ETp calculation.

**R2C6:** Model Performance Metrics: The performance metrics (e.g., $R^2$, RMSE) for the RF regressor and XGBoost model are not provided. These metrics are crucial for evaluating model performance. Additionally, the rationale for choosing RF over XGBoost should be explained.

**Reply: DONE.** Thank you for pointing this out, and we apologize for the oversight. As per your suggestion, we have checked the performance metrics of the training and validation sets. Despite using grid search to optimize model parameters, the large discrepancy between training and validation performance indicated overfitting in the original models. To address this, we re-ran the machine learning analysis with the following three major modifications:

**(1) Input variable adjustment:** Following the Editor's suggestion, we removed the

weighted average soil moisture variables (0–100 cm and 0–289 cm) and instead introduced root-zone soil moisture based on root depth estimates from Stocker et al. (2023).

**(2) Overfitting control:** We expanded the hyperparameter search space for both the RF and XGBoost regressors to reduce model complexity and implemented "Early Stopping" to prevent overfitting.

**(3) Improved modeling strategy:** We divided the data into multiple subsets based on site characteristics and time scales. Each subset was calibrated using all 17 input variables. We then aggregated the variable importance rankings across subsets to identify robust predictors (Figure 4 in the main text; Figure S3 in the Supplement).

Despite these efforts, strong collinearity among similar input variables still led to noticeable differences between training and validation performance, with XGBoost performing better on the validation set (Tables S2 and S3). Therefore, for partial dependence analysis, we used the SHAP model based on the XGBoost regressor with only the top eight ranked variables as input. To enhance the robustness of the PDPs, we conducted **30 iterations with different random seeds and averaged the results** (Figure 5 in the main text; Figures S4 and S5 in the Supplement).

Model performance is reported as follows (Tables S2 and S3 in the Supplement):

**Table S2. Performance metrics for the RF and XGBoost regressor models.**

It presents performance metrics ($R^2$, RMSE, MAE) of RF and XGBoost regressor models across 8-day, 16-day, and monthly temporal scales. Results are reported for both training and validation sets (70:30 split). Higher $R^2$ values (closer to 1) and lower RMSE/MAE values (approaching 0) indicate superior model performance. It reveals that XGBoost regressor model demonstrates higher accuracy than RF on the validation set, with a smaller accuracy discrepancy between the training and validation sets, thereby demonstrating superior generalization capability of the XGBoost model.

| Classification | Metrics | 8day | | | | 16day | | | | Month | | | |
|---|---|---|---|---|---|---|---|---|---|---|---|---|---|
| | | Training | | Validation | | Training | | Validation | | Training | | Validation | |
| | | RF | XGBoost | RF | XGBoost | RF | XGBoost | RF | XGBoost | RF | XGBoost | RF | XGBoost |
| Cropland | $R^2$ | 0.93 | 0.90 | 0.74 | 0.77 | 0.93 | 1 | 0.67 | 0.71 | 0.95 | 0.89 | 0.69 | 0.65 |
| | RMSE | 0 | 0.001 | 0.002 | 0.002 | 0 | 0 | 0.002 | 0.002 | 0 | 0 | 0.001 | 0.002 |
| | MAE | 0.017 | 0.022 | 0.034 | 0.032 | 0.013 | 0.003 | 0.034 | 0.032 | 0.010 | 0.017 | 0.028 | 0.029 |
| Forest | $R^2$ | 0.93 | 0.93 | 0.67 | 0.74 | 0.92 | 0.86 | 0.63 | 0.62 | 0.93 | 0.98 | 0.45 | 0.51 |
| | RMSE | 0 | 0.001 | 0.004 | 0.003 | 0 | 0.001 | 0.002 | 0.002 | 0 | 0 | 0.002 | 0.002 |
| | MAE | 0.020 | 0.021 | 0.042 | 0.039 | 0.016 | 0.024 | 0.034 | 0.035 | 0.011 | 0.006 | 0.035 | 0.033 |
| Grassland | $R^2$ | 0.93 | 0.98 | 0.63 | 0.66 | 0.92 | 0.83 | 0.59 | 0.59 | 0.93 | 1 | 0.54 | 0.58 |
| | RMSE | 0 | 0 | 0.001 | 0.001 | 0 | 0 | 0.001 | 0.001 | 0 | 0 | 0.001 | 0.001 |
| | MAE | 0.010 | 0.005 | 0.021 | 0.021 | 0.009 | 0.013 | 0.020 | 0.020 | 0.007 | 0.001 | 0.019 | 0.019 |
| NonDryland | $R^2$ | 0.97 | 0.92 | 0.70 | 0.72 | 0.95 | 0.97 | 0.61 | 0.61 | 0.86 | 0.85 | 0.52 | 0.52 |
| | RMSE | 0 | 0 | 0.001 | 0.001 | 0 | 0 | 0.001 | 0.001 | 0 | 0 | 0.001 | 0.001 |
| | MAE | 0.011 | 0.015 | 0.028 | 0.027 | 0.009 | 0.006 | 0.023 | 0.023 | 0.010 | 0.011 | 0.021 | 0.021 |
| Dryland | $R^2$ | 0.98 | 0.96 | 0.83 | 0.85 | 0.97 | 1 | 0.80 | 0.84 | 0.97 | 1 | 0.70 | 0.69 |
| | RMSE | 0 | 0 | 0 | 0 | 0 | 0 | 0 | 0 | 0 | 0 | 0 | 0 |
| | MAE | 0.006 | 0.009 | 0.018 | 0.016 | 0.006 | 0.003 | 0.016 | 0.015 | 0.006 | 0 | 0.017 | 0.017 |
| SemiHumid | $R^2$ | 0.95 | 0.93 | 0.74 | 0.75 | 0.95 | 0.94 | 0.73 | 0.73 | 0.96 | 0.86 | 0.60 | 0.64 |
| | RMSE | 0 | 0 | 0.001 | 0.001 | 0 | 0 | 0.001 | 0.001 | 0 | 0 | 0.001 | 0.001 |
| | MAE | 0.012 | 0.015 | 0.026 | 0.026 | 0.011 | 0.012 | 0.026 | 0.026 | 0.008 | 0.015 | 0.023 | 0.023 |
| SemiArid | $R^2$ | 0.97 | 0.98 | 0.80 | 0.82 | 0.96 | 0.99 | 0.79 | 0.81 | 0.96 | 0.98 | 0.56 | 0.66 |
| | RMSE | 0 | 0 | 0.001 | 0.001 | 0 | 0 | 0.001 | 0.001 | 0 | 0 | 0.001 | 0.001 |
| | MAE | 0.010 | 0.007 | 0.025 | 0.024 | 0.011 | 0.005 | 0.025 | 0.023 | 0.008 | 0.006 | 0.028 | 0.025 |
| Arid | $R^2$ | 0.98 | 0.97 | 0.92 | 0.92 | 0.93 | 0.98 | 0.80 | 0.84 | 0.98 | 0.99 | 0.80 | 0.82 |
| | RMSE | 0 | 0 | 0 | 0 | 0 | 0 | 0 | 0 | 0 | 0 | 0 | 0 |
| | MAE | 0.005 | 0.007 | 0.011 | 0.011 | 0.007 | 0.005 | 0.014 | 0.013 | 0.004 | 0.003 | 0.011 | 0.011 |

**Table S3. Performance metrics for the XGBoost regression model after variable pre-screening.**
The input variables for parameter sets in the table were configured as follows:
- R0: svm0-7, Rn, T, VPD, U, P, $CO_2$, and LAI
- R1: replaced svm0-7 with Prec while retaining all other variables from R0

| Classification | Metrics | 8day | | 16day | | month | |
|---|---|---|---|---|---|---|---|
| | | Training | Validation | Training | Validation | Training | Validation |
| R0 | $R^2$ | 0.82 | 0.64 | 0.85 | 0.62 | 0.86 | 0.62 |
| | RMSE | 0 | 0.001 | 0 | 0.001 | 0 | 0 |
| | MAE | 0.017 | 0.023 | 0.012 | 0.019 | 0.009 | 0.015 |
| R1 | $R^2$ | 0.82 | 0.65 | 0.93 | 0.58 | 0.85 | 0.52 |
| | RMSE | 0 | 0.001 | 0 | 0.001 | 0 | 0.001 |
| | MAE | 0.018 | 0.024 | 0.009 | 0.021 | 0.010 | 0.018 |

The corresponding results are shown below:

[Figure]

**Figure R10 (Figure 4 in the main text, Line 310). Identification of the key external drivers of evapotranspiration stress index (ESI).** (a) Matrix plot employs bubble size to represent the absolute values of SHAP, with a color gradient from blue to red denoting negative and positive effects, respectively. Numerical annotations within each bubble denote the rank of importance for the variables, with higher values signifying greater influence. These ranks are obtained from an explainable machine learning utilizing XGBoost regressors. (b) Boxplot provides a statistical representation of the variable rankings across varying underlying surface conditions (corresponding to the row it's in), where the black line denotes the mean ranking and the red dot signifies the median ranking. The analysis incorporates vegetation factors such as FVC, EVI, NDVI, kNDVI, and LAI; meteorological factors including atmospheric pressure (P), carbon dioxide (CO2) concentration, wind speed (U), and VPD; energy factors encompass air temperature (T) and net radiation (Rn); and water supply factors, which account for precipitation (Prec) and the soil volumetric water content (svm) across various soil layer depths, with subscripts specifying the depth range, with 0-7 indicating the soil layer from 0 to 7cm. Inputs to the model are the anomalies of all aforementioned variables.

[Figure]

**Figure R11 (Figure 5 in the main text, Line 325). Importance and partial dependence of external driving factors influencing ESI.** (a) Relative importance of hydrologic, energy, climatic, and vegetation factors in explaining ESI, with each category encompassing multiple indicators. The significance of these indicators is visually encoded by the color scheme of the bars matching their labels on the left. (b) Pie chart illustrates the proportion of the four categories described in (a). (c-j) Partial dependence graphs correspond to the following variables, in order:P, U, CO2, VPD, T, Rn, LAI, and soil moisture content for the top 0-7 cm. The thickened lines indicate the average effects, and the light lines around them indicate 30 random incidents from the data sets. This figure is based on an 8-day data series and generated through a SHAP model employing a XGBoost regressors, which processes the anomalies of the variables as inputs.

[Figure]

**Figure R12 (Figure S3 in Supplement). Identification of the key external drivers of ESI.** Same as Fig. 4, but applying the RF regressor. It is consistent with the overall results identified by the XGBoost classifier, except that it fails to distinctly differentiate between surface soil moisture and precipitation.

[Figure]

**Figure R13 (Figure S4 in Supplement). Importance and partial dependence of external drivers influencing ESI.** Same as Fig. 5, but here examines the 16-day temporal resolution. The hierarchy of importance is as follows: water, climate, energy, and then vegetation. Compared to the 8-day temporal resolution findings, the importance of water and vegetation factors remains virtually unchanged, with svm0-7, VPD, Rn and LAI as the most influential variables within their respective categories.

[Figure]

**Figure R14 (Figure S5 in Supplement). Importance and partial dependence of external drivers influencing ESI.** Same as Fig. 5 & S4, but here analyze monthly series data. The order of importance is water > climate > vegetation > energy. Compared to the results from the 8-day and 16-day temporal results, the significance of water is amplified (with an increase to 0.56), whereas the contributions of vegetation and energy factors are diminished, with the latter being particularly affected. Factors such as svm0-7, VPD, LAI, and Rn continue to be the most influential within their respective categories, aligning with observations from the other temporal scale. The dependency direction of the pivotal factor aligns with that of the other two temporal dimensions.

Relevant text now reads:

Line 183-184 in **2.2 Identification of the key drivers** (Materials and Methods):

*"Furthermore, regularization parameters were integrated alongside the "Early Stopping" mechanism to mitigate overfitting risks and rigorously control model complexity."*

Line 187-188 in **2.2 Identification of the key drivers** (Materials and Methods):

*"To enhance the robustness of PDPs, we performed 30 iterations with varying random seeds and averaged the resulting estimates."*

Line 278-285 in **3.2 Soil moisture as the primary external driver** (Results):

*"We assessed the relative importance of water supply, climatic, energy, and vegetation factors on the ESI across various temporal scales and regional characteristics through an interpretable machine learning model utilizing RF regressor and XGBoost regressor. Table S2 presents the performance metrics of two models. Despite extensive efforts to prevent overfitting through parameter optimization (Method 2.2), the relatively low validation accuracy still reveals SHAP*

*model's susceptibility to misallocating feature importance among highly correlated variables. To address this, we generated multiple training subsets through categorical divisions of temporal scales (8-day, 16-day, monthly) and spatial partitions (drought gradient and land cover types: cropland, forest, grassland). This approach enabled us to obtain diverse importance rankings across China's regions and derive statistically robust importance hierarchies through distribution analysis (Fig. 4, S3)."*

Line 289-292 in Section **3.2 Soil moisture as the primary external driver** (Results) explains the rationale for selecting the XGBoost model:

*"Similar importance ranking distributions were observed using RF regressor (Fig. S3b). Comparatively, the XGBoost regressor exhibited enhanced stability across multiple subsets, with minimal divergence between training and validation performance metrics (Table S2), prompting its selection for subsequent importance quantification analysis."*

Line 298-301 in **3.2 Soil moisture as the primary external driver** (Results) :

*"Following collinear variable filtering, we identified svm0-7, Rn, T, VPD, U, P, CO2, and LAI as critical external drivers of evaporative stress (parameter set R0). Given precipitation's secondary ranking to svm0-7, we established scenario R1 substituting precipitation to examine hydrological driver divergence (Appendix C). SHAP model calibration across temporal scales revealed ESI response patterns for both variable sets. Reduced multicollinearity enhanced model stability (Table S3)."*

**R2C7:** Multicollinearity and SHAP Values: Despite efforts to address multicollinearity, the SHAP values for soil moisture appear contradictory to precipitation (Figure 5). Given that these variables are generally positively correlated, the authors should explain this discrepancy.

**Reply: DONE.** Thank you for pointing this out. The figure in question (now Figure 4) was intended for preliminary screening of influential variables. The input variables (17 in total) exhibit strong multicollinearity due to representing similar physical meanings. This multicollinearity can distort SHAP value interpretations, causing inaccurate attribution of importance among highly correlated variables. We emphasize that the SHAP values in this figure are biased, as also evidenced by the large accuracy gap between the training and validation sets shown in Table S2, suggesting limited reliability of the SHAP results in this step.

Nevertheless, this step serves to identify key variables. To obtain more robust importance rankings, we categorized the data and used multiple time scales. The boxplots in Figure 4 summarize the distribution of rankings across spatial categories and time scales, with medians highlighted.

For the subsequent PDP analysis (Figure 5), only the top 8 variables identified from the previous step were used. The improved consistency between training and validation accuracy in Table S3 supports the model's enhanced reliability.

The models and results related to this revision are detailed in our response to Comment 6.

In response to your suggestion, we have added Appendix C, which further explores the importance and partial dependence of ESI on precipitation and soil moisture based on SHAP analysis. Please refer to Lines 663–686 for the added content.

**"Appendix C: Comparative impacts of precipitation and surface soil moisture on ESI**

*Given that precipitation exhibited the second-highest statistical rank among parameters (surpassed only by svm0-7), we developed a comparative Parameter Set R1 using precipitation anomalies to investigate the distinct influences of precipitation and soil moisture on the ESI. SHAP model analysis across multiple temporal scales in China revealed contrasting feature importance patterns between parameter sets (see Table S3 for model performance metrics): R1 displayed broader contribution distributions indicating balanced parameter interactions, while R0 prioritized dominant parameters (svm0-7, Rn). Both configurations consistently showed Rn, VPD, and T negatively regulating ESI, contrasting with positive regulation by LAI and U (Fig. C1). Notably, Prec and svm0-7 synergistically enhanced ESI through moisture supply mechanisms (Fig. C1, C2, 5j, S4j-S5j), consistently ranking as top predictors. In R1, Prec showed persistent secondary influence to VPD, whereas SVM0-7 dominated feature importance at both 16-day and monthly scales (Fig. C1b-c). Diverging from soil moisture's linear relationship, precipitation anomalies induced a biphasic "plateau-ramp" ESI response, requiring anomalies exceeding -5 to 0mm for stress mitigation. Conversely, svm0-7 exhibited acute sensitivity with near-vertical monthly response gradients (Fig. S5j), indicating immediate drought relief from minimal moisture replenishment (-0.02 to -0.01 $m^3 \cdot m^{-3}$) under extreme deficits. Our deseasoned-detrended anomaly analysis inherently accommodates negative values. Subsequent research should prioritize identifying precise response thresholds and turning points."*

[Figure]

**Figure R15 (Figure C1 in Supplement, Line 680). Multi-temporal beeswarm plots between Parameter Set R0 (a-c) and R1 (d-f).** Parameter set R0 includes: svm0-7, Rn, T, VPD, U, P, CO₂, LAI, with R1 substituting svm0-7 with Prec. Red denotes high feature values with substantial impacts on ESI. High feature values distributed along the SHAP positive half-axis signify positive contributions to ESI. All features are arranged in descending order of contribution magnitude, where wider x-axis distributions reflect stronger model impacts.

[Figure]

**Figure R16 (Figure C2 in Supplement, Line 685).** Precipitation PDPs under Parameter Set R1 across temporal scales, with soil moisture counterparts shown in Figs. 5j, S4j, and S5j.

**R2C8:** Group Differences: Figure 5 groups samples into cropland, forest, and grassland, but differences among these groups are not adequately discussed. For instance, grassland shows different behavior compared to cropland and forest for svm0-7. The authors should explore the possible physical mechanisms behind these differences.

**Reply: DONE.** Thank you for pointing this out, our original explanation was indeed unclear. The grouping in Figure 5 (now Figure 4) was intended to reduce potential SHAP value allocation errors caused by multicollinearity among variables. By creating subsets, we were able to perform multiple rankings and derive statistically robust importance scores for identifying key drivers. Further details and clarification can be found in our responses to Comments 6 and 7.

In addition, following your suggestion, we have added a grouped analysis of the multi-year mean and trends of the core result — the time-varying sensitivity — in Figure R17 (see also Figure 8 in the main text), with corresponding descriptions included in Section 3.3.

[Figure]

**Figure R17 (Figure 8 in main text, Line 418). Sensitivity comparisons between MDLM and DLM models across regional land types.** (a-c) Distributions of sensitivity parameters $\theta$ (ESI to svm, VPD, and LAI) in dryland, nondryland, forest, cropland, and grassland. Boxplots show interquartile ranges (25th–75th percentiles, boxes), medians (central lines), means (dots), and data ranges (5th–95th percentiles, whiskers). Corresponding probability density curves are plotted alongside. Top asterisks indicate t-test significance between different groups: three asterisks (*) denote p < 0.001, N.S. = non-significant differences.

Relevant text reads:

Line 367-369:

*"Additionally, $\theta_{svm}$ demonstrates considerable variation among different land-use types, with forest shows the strongest sensitivity and the most drastic changes, followed by cropland and grassland (MLDM: 4.27>3.60>1.17, Fig.8ad)."*

Line 380-384:

*"Moreover, $\theta_{VPD}$ in arid regions is decreasing (-0.89% decade$^{-1}$, p < 0.01), in contrast to an increasing trend observed in humid regions (3.36% decade$^{-1}$, p < 0.001), so that the discrepancy between dryland and nondryland is anticipated to widen (Fig.8be). Notably, MLR indicates an even more pronounced divergence in trends between dryland and nondryland regions (-2.41% vs. 6.58% decade$^{-1}$, p < 0.001, Fig.8e)."*

Line 396-400:

*"Figure 8 further illustrates that the values and trends of $\theta_{svm}$, $\theta_{VPD}$, and $\theta_{LAI}$, as estimated by MLDM and MLR, display statistically significant differences (p < 0.001) across nearly all climate zones and vegetation types, underscoring substantial methodological discrepancies. Within the same land use categories, the MLR model consistently demonstrates more dispersed sensitivity trend distributions and greater intra-group variability (Fig.8)."*

**R2C9:** Soil Moisture Sensitivity Changes: The reasoning behind the changes in soil moisture sensitivity is not convincing. The authors attribute these changes to anomalies in the year 2016 (Lines 362-365). It is unclear how a single-year anomaly can explain long-term changes.

**Reply: Corrected.** This point was also noted by another reviewer, and we fully agree. Our initial explanation was inaccurate. We reanalyzed the results based on the revised figure. As both methods consistently indicate a peak in ESI sensitivity to soil moisture around 2015, we believe our findings are robust. Furthermore, we identified supporting evidence in the study by Tang et al. (2025), which reported a similar temporal pattern—an initial increase followed by a subsequent decline—in vegetation sensitivity to soil drought. By employing 12 TRENDY models within a controlled experimental framework, they attributed this transition primarily to the influence of climate change. Their findings lend support to our results and offer a plausible explanation for the observed pattern. Corresponding revisions have been made in the main text to reflect this connection.

Relevant text reads (Line 349-355):

*"Soil moisture stands out as the paramount and most sensitive factor, with its sensitivity demonstrating a significant upward trend in both models, albeit with quantitative differences (Fig. 6b). According to the MDLM results, $\theta_{svm}$ increased by 9.49% over the past two decades (0.03±0.01 yr$^{-1}$, p < 0.001), peaking in 2016, which represents a 13% rise compared to the stable level of the previous decade. In contrast, the MLR model, while producing a slightly lower mean $\theta_{svm}$ (2.36 vs. 2.98), exhibited a more pronounced growth trend (0.05±0.01 yr$^{-1}$, p < 0.001), with the peak occurring earlier in 2014. This shift in peak timing is an inherent*

*consequence of the moving average process. As observed in similar studies, Tang et al. (2025) reported analogous evolutionary patterns and attributed this phenomenon primarily to climatic drivers."*

Tang, J., Niu, B., Fu, G., Peng, J., Hu, Z., and Zhang, X.: Shifted trend in drought sensitivity of vegetation productivity from 1982 to 2020, Agricultural and Forest Meteorology, 362, 110388, https://doi.org/10.1016/j.agrformet.2025.110388, 2025.

**R2C10:** Key Findings Summary: The key points of the paper are not clearly summarized. I recommend the authors provide a concise summary of their key findings at the end of the paper to enhance clarity.

**Reply: DONE.** We have added a Conclusion section at the end of the manuscript (Line X) to concisely highlight the main findings of this study. In addition, new content and insights emerged during the revision process. Accordingly, we have reorganized and refined the structure of Section 4 (Discussion) to improve clarity and coherence. We hope these revisions enhance the overall readability of the manuscript and provide meaningful new perspectives on the evolving relationships among complex variables within ecosystems. The newly added Conclusion section (Line 535-547) reads as follows:

**"5 Conclusions**

*Our study introduces a memory-driven dynamic linear model (MDLM) that integrates the "dry-get-drier" legacy effects to evaluate time-specific sensitivities of evapotranspiration stress drivers in China under rapid vegetation greening. Using the Evapotranspiration Stress Index (ESI) as a proxy for ecosystem water-atmosphere-vegetation equilibrium, we documented a 4.74% intensification of ET stress across mainland China from 1950 to 2020. Soil moisture sensitivity was identified as the dominant driver, showing a 9.49% increase during 2001–2020. Enhanced vegetation greening exhibited stronger coupling with elevated soil moisture sensitivity but reduced VPD sensitivity, reflecting greater susceptibility to soil drought concurrent with improved resilience to atmospheric aridity. Compared to the conventional moving-window multiple linear regression (MLR) method, the MDLM framework increased the coefficient of determination ($R^2$) by 44.26%, significantly improving sensitivity estimates for critical drivers. Methodologically, this work underscores the need to critically re-evaluate trend overestimation inherent in traditional MLR methods. Our findings advance mechanistic understanding of complex regional ecosystem dynamics and offer guidance for steering greening strategies toward a more stable equilibrium among water, atmosphere, and vegetation."*

Comparison diagram of the overall structural adjustments of the manuscript:

[Figure]

**Figure R18. Comparison of manuscript structure before and after revision.**

We sincerely thank you once again for your constructive and insightful comments, which have significantly improved the quality of our manuscript and also led to some unexpected and valuable findings. Should any parts of our response remain unclear, please do not hesitate to let us know. We greatly appreciate your continued guidance.

---

## Author Response (AR2)

**Response to Editor**

Dear Prof. Gao,

We greatly appreciate your timely handling of our manuscript. In response to the reviewer's suggestion to improve the readability, we have made the following changes:

a) We simplified the language throughout the manuscript, including in Appendices A, B, and C, using more concise phrasing while maintaining the original meaning.

b) Some descriptions in the results section were moved to the methods or removed to emphasize the key findings.

All changes are highlighted in blue in the revised version. Thank you for your support, and we look forward to the successful publication.

Best,
Yuan Liu (on behalf of all co-authors)

**Response to Reviewer #2**

**General Comments:** The authors have made great efforts to revise the manuscript and have effectively addressed most of my concerns. The structure and quality of the presentation have improved considerably.

**Reply:** We greatly appreciate your recognition, as well as your previous review and guidance, which have been very valuable to us.

**R2C1:** 1) Some paragraphs are overly long and could be more concise. I recommend limiting each paragraph to about 10 lines by either splitting or simplifying the descriptions. For example, in Section 2.2, consider dividing the first paragraph into two: one focusing on the driver candidates and the other on the ML methods.

**Reply: DONE.** In response to your feedback, we have made two key changes to improve the manuscript's clarity and readability: First, we refined the language throughout the manuscript and Appendices A, B, and C, with all changes highlighted in blue. Additionally, following your guidance and examples, we divided several sections into shorter paragraphs to keep each paragraph within 10 lines.

The paragraph divisions are at: Line 164, Line 243, Line 387, Line 438, and Line 465.

Additionally, we have added the heading "2.4 Other Statistical Analysis" in the methods section to better organize the content (Line 224).

**R2C2:** 2) The contents in the results should appear in the order from important ones to less important ones. For instance, in section 3.2, the authors could firstly describe their findings, and then the robustness of the methods (or just move them into method sections). Similarly, in section 3.3, the comparison of the two models is less important than the findings based on the current model, and should appear later, possibly only in the

discussion section.

**Reply: DONE.** We greatly appreciate your detailed guidance. Following your suggestion, we have moved the relevant content from Section 3.2 to Methods 2.2. Regarding the model comparison analysis in Section 3.3, we found it to be repetitive with the discussion in Section 4.1, so we have removed it.

Relevant text reads (Line 170-175):

> *"Table S2 presents the performance metrics of two models. Despite above efforts to prevent overfitting, the relatively low validation accuracy still reveals SHAP model's susceptibility to misallocating feature importance among highly correlated variables. To address this, we generated multiple training subsets through categorical divisions of temporal scales (8-day, 16-day, monthly) and spatial partitions (drought gradient and land cover types: cropland, forest, grassland). This approach enabled us to obtain diverse importance rankings across China's regions and derive statistically robust importance hierarchies through distribution analysis (Fig. 4, S3)."*

We truly appreciate your insightful guidance and invaluable contributions to improving our manuscript. Best regards.